# Efficient generation of germline chimeras in a non-rodent species using rabbit induced pluripotent stem cells

Hong-Thu Pham[1,8], Florence Perold[1,8], Yannicke Pijoff [1,8], Nathalie Doerflinger[1], Sylvie Rival-Gervier[1], Maëlle Givelet[1], Anaïs Moulin[1], Manon Ressaire [1], Emilie Da Silva Fernandes[1], Valeska Bidault[1,2], Luc Jouneau[3,4], Véronique Duranthon[3,4], Florence Wianny[1], Bertrand Pain [1], Ingrid Plotton[1,2], Thierry Joly[5,6], Marielle Afanassieff [1,7,9] ✉, Pierre Savatier [1,7,9] ✉ & Nathalie Beaujean [1,9] ✉

Pluripotent stem cells have long been used to produce knockout mice via germline chimera technology. However, aside from the rat, this approach has not been successfully applied to other mammals. Here, we demonstrate that rabbit induced pluripotent stem cells (iPSCs) can be reprogrammed using KLF2, ERAS and PRMT6, enabling them to efficiently colonize embryos. These chimeric embryos can develop into fetuses and newborn rabbits, with iPSCs contributing up to 100 % to certain organs. Notably, female rabbits generated through this method are healthy and transmit the iPSC genome to their offspring with a high efficiency, demonstrating germline chimerism. This advancement establishes a foundation for developing rabbit models of human disease with complex genetic traits.

Germline chimerism is the ability of embryonic stem cells (ESCs) and induced pluripotent stem cells (iPSCs) to contribute to the formation of all tissues and organs in both the fetus and the adult organism when introduced into blastocysts or aggregated with morulae. To date, germline chimerism has been documented exclusively in mice and rats and is widely regarded as the definitive test for pluripotency. It provides unequivocal evidence that ESCs and iPSCs can generate fully functional cells across all tissues and organs[1,2]. By leveraging genome editing, ESCs and iPSCs have greatly advanced studies of mouse development and facilitated the creation of mouse models of human genetic diseases. It is generally accepted that only ESCs and iPSCs in the naïve state of pluripotency can colonize preimplantation embryos and subsequently differentiate into all lineages, enabling germline chimera formation[3]. In contrast, pluripotent cells self-renewing in the primed state of pluripotency, known as epiblast stem cells (EpiSCs)[4,5], cannot form chimeras when introduced into preimplantation embryos.

The development of germline chimera technology in domestic animals is highly anticipated, as it could lead to the generation of new models for human diseases[6,7]. However, this technology is still in its early stages. In rabbits, available ESC and iPSC lines exhibit key characteristics of primed pluripotency[8]. When cultured under conditions suitable for maintaining naïve pluripotency in rodents or primates, rabbit ESCs and iPSCs proliferate slowly, display spontaneous differentiation, and become chromosomally unstable[9]. Despite these challenges, chimeric conceptuses at 10 days of development were achieved following the injection of naïve-like iPSCs into morulae[10]. These rabbit iPSCs overexpressed human KLF2 and KLF4 and were

[1]Univ Lyon, Université Lyon 1, INSERM, Stem Cell and Brain Research Institute U1208, INRAE USC 1361, F-69500 Bron, France. [2]Hospices Civils de Lyon, F-69000 Lyon, France. [3]Université Paris-Saclay, UVSQ, INRAE, BREED, 78350 Jouy-en-Josas, France. [4]Ecole Nationale Vétérinaire d'Alfort, BREED, 94700 Maisons-Alfort, France. [5]ISARA-Lyon, F-69007 Lyon, France. [6]VetAgroSup, UPSP ICE, F-69280 Marcy l'Etoile, France. [7]PrimaStem Platform, Univ Lyon, Université Lyon 1, INSERM, Stem Cell and Brain Research Institute U1208, Lyon, France. [8]These authors contributed equally: Hong-Thu Pham, Florence Perold, Yannicke Pijoff. [11]These authors jointly supervised this work: Marielle Afanassieff, Pierre Savatier, Nathalie Beaujean. ✉e-mail: marielle.afanassieff@inserm.fr; pierre.savatier@inserm.fr; nathalie.beaujean@inserm.fr

adapted for propagation in medium supplemented with leukemia inhibitory factor (LIF) and serum. However, the rate of chimerism was very low, with fewer than 1 in 1000 cells originating from iPSCs. In pigs, chimeric fetuses at 4 to 7 weeks were reported following the injection of naïve-like iPSCs into morulae or blastocysts[11,12]. In bovine, 3-month-old chimeric fetuses were achieved through the aggregation of naïve-like iPSCs with morulae[13]. As with rabbit, pig and bovine chimeric fetuses exhibited very low levels of chimerism, with no evidence of germline colonization.

A critical requirement for generating germline chimeras is the ability of pluripotent cells to robustly self-renew in the naive state. In mice, this is typically achieved using culture conditions containing LIF, MEK inhibitor PD0325901, and GSK3 inhibitor CHIR99021, known as 2iLIF[3,14,15]. Germline chimerism has also been achieved by reprogramming EpiSCs through the overexpression of specific transcription factors, including Klf4, Nr5a1, and Nr5a2, in 2iLIF culture conditions[16,17]. The first naïve-like human ESCs were generated by reprogramming conventional primed pluripotent stem cells through transient overexpression of OCT4, KLF2, KLF4, and NANOG in 2iLIF or 3iLIF (i.e., 2iLIF + PKC inhibitor Gö6983)[18,19].

To pursue a similar objective in rabbits, we conducted an unbiased screening of a cDNA library encoding up to 36 pluripotency factors. In this study, we identify genes whose overexpression in rabbit iPSCs enables the formation of chimeric embryos, fetuses, and newborns, with significant iPSC contributions in all major organs, including the germline. Female rabbits generated through this method are healthy and transmit the iPSC genome to their offspring with high efficiency. Moreover, we characterize the transcriptomic signature of chimeric-competent iPSCs and identify candidate regulatory pathways associated with embryo colonization.

## Results

### The VALGöX culture regimen steers iPSCs toward the naïve state

The rabbit iPSC (rbiPS) line B19 used throughout the study was previously established from rabbit ear fibroblasts using retroviral vectors that encode human OCT4, SOX2, KLF4, and c-MYC. All four transgenes were silenced after reprogramming[8]. B19 iPSCs are commonly cultured in basal medium supplemented with FGF2 and knockout serum replacement (KOSR) factors, or KF medium, on feeders (referred to as B19_KF[8,10]). When attempts were made to grow B19_KF cells in media originally described for naive human and rhesus monkey ESCs/iPSCs (i.e., ENHSM[20], 5iLA[21], t2iLGöY[19], 4i/L/b[22], and LCDM[23]), the cells either died or differentiated (Supplementary Fig. 1a). However, when cultured in murine embryonic fibroblasts (MEF)-conditioned N2B27 basal media supplemented with activin A, human LIF, Vitamin C, protein kinase C inhibitor Gö6983, and tankyrase inhibitor XAV939 (hereafter called VALGöX), B19 cells displayed progressive morphological and molecular changes. After 48 h in VALGöX (referred to as B19_VAL), they formed more compact colonies (Fig. 1a, b). Immunofluorescence analysis also revealed higher expression of two naïve markers, DPPA5 and OOEP[24], increased levels of the permissive histone mark H3K14ac, decreased levels of the repressive histone marks H3K9me3, and decreased levels of 5-methyl-Cytosine (5mC), suggesting the onset of a transition from primed to naïve-like pluripotency (Fig. 1b). Moreover, transcriptome analysis showed reduced expression of rabbit primed and gastrulation markers and increased expression of rabbit formative markers[24] (Fig. 1c). However, B19_VAL cells lacked certain key characteristics of naïve PSCs. Specifically, immunostaining for H2AK119ub and H3K27me3, two histone post-translational modifications associated with X chromosome inactivation in mice and humans[25], exhibited nuclear foci in 69% (H2AK119ub) and 80% (H3K27me3) of B19_VAL cells, and in 78% (H2AK119ub) and 75% (H3K27me3) of B19_KF cells (Supplementary Fig. S1b, c). This suggests the presence of an inactive X chromosome in the majority of both B19_KF and B19_VAL cells. Additionally, both B19_VAL and B19_KF cells demonstrated limited capacity

for epiblast contribution after injection into early morula-stage embryos (E2.8) (Supplementary Fig. S1d). In summary, these findings suggest that the VALGöX culture regimen promotes the development of rabbit iPSCs towards a more immature pluripotent state, but does not achieve genuine naïve-state reprogramming. After 14 days in VALGöX, B19_VAL_14d cells exhibited morphological differentiation (Fig. 1d). Transcriptomic analysis revealed a decline in the expression of naïve, formative, and primed pluripotency markers, alongside an upregulation of lineage markers, confirming the initiation of differentiation (Fig. 1e). The VALGöX culture regimen therefore fails to support B19 cell long-term self-renewal.

To monitor the transition from primed to naïve pluripotent states and to identify rare cells undergoing this conversion in B19 cell cultures, we developed a dual fluorescent naïve pluripotency reporter. *DPPA2* was selected as a marker of naïve pluripotency in rabbits due to its expression profile in preimplantation embryos. Single-cell RNA sequencing analyses indicated that *DPPA2* is expressed in all pluripotent cells of day 3 (early-blastocyst stage) and day 4 (mid-blastocyst stage) rabbit embryos, with expression declining from day 5 onwards, suggesting downregulation during the transition from naive to primed pluripotency (Supplementary Fig. S1e)[24]. Starting from B19 cells that expressed several copies of the GFP transgene–driven by the EF1α promoter–obtained through multiple infection with a lentiviral vector[8], we integrated monomeric Kusabira Orange (*mKO2*) downstream of the *ATG* codon of *DPPA2* (Supplementary Fig. S1f). The resulting B19-EF1α$^{GFP}$-DPPA2$^{mKO2}$ cells were further transfected with *pEOS$^{tagBFP}$*, a reporter plasmid containing blue fluorescent protein (tagBFP) under the naïve-specific distal enhancer of mouse *Pou5f1/ Oct4*[26] (Supplementary Fig. S1g). A clone of B19-EF1α$^{GFP}$-DPPA2$^{mKO2}$-EOS$^{tagBFP}$ cells, named NaiveRep, was selected for further study. NaiveRep exists in two forms: NaiveRep_KF cells and NaiveRep_VAL, the latter obtained after culturing NaiveRep_KF cells in VALGöX medium for 48 h (Fig. 1f). Interestingly, NaiveRep_VAL cells expressed the fluorescent reporter EOS-TagBFP at a slightly higher level than NaiveRep_KF cells, while no difference in mKO2 fluorescence levels was observed between the two cell types (Fig. 1g). Overall, these findings confirm the ability of the VALGöX culture medium to initiate conversion to naïve pluripotency, but they also highlight its inability to complete the process.

### cDNA screening identifies genes supporting the naïve state

A library of 25 human cDNAs was generated using the W10 simian immunodeficiency virus (SIV)-based backbone lentiviral vector. The selected cDNAs encode transcriptional factors (*NANOG, SOX2, KLF2, KLF4, KLF5, KLF12, ESRRB, ESRRG, GBX2, TFCP2L1, NR5A2, SALL4, MYC,* and *GFI1*), a signaling molecule (*ERAS*), histone modifying enzymes, and chromatin modifiers (*PRMT6, KHDC1, KAT2B, KDM4D, SUV39H1, BMI1,* and *GASC1*), a cell-cycle protein (*CCNE1*), and viral oncogenes, including SV40 Large T antigen (AgT) and adenovirus E1A-12S. Many of these selected genes have been demonstrated to support naïve pluripotency or facilitate the resetting to naïve-like features when overexpressed in mice and humans[16,17,19,27–41]. Our RNA-seq analyses of rabbit embryos confirmed that all but two genes (*GFI1* and *ERAS*) were expressed in the ICM/early epiblast of day 3 and 4 rabbit blastocysts[24] (Supplementary Fig. S2a). The relative titers of the 25 viruses were estimated by serial dilution (Supplementary Fig. S2b). Four additional lentiviral vectors expressing GFP, mKO2, tagBFP, and Katushka, respectively, were used to calibrate the multiplicity of infection (MOI) (Supplementary Fig. S2c, d).

A total of $2.4 \times 10^5$ NaiveRep cells were transduced with the cDNA library using a MOI of 50, and subsequently cultured for 7 days (Fig. 2a). Nearly 5000 colonies were examined under epifluorescence, and 27 exhibiting increased mKO2- and/or tagBFP-associated fluorescence were selected and amplified for further analysis (subsequently termed NaiveRep_KF clones). Flow cytometry analysis of tagBFP and mKO2

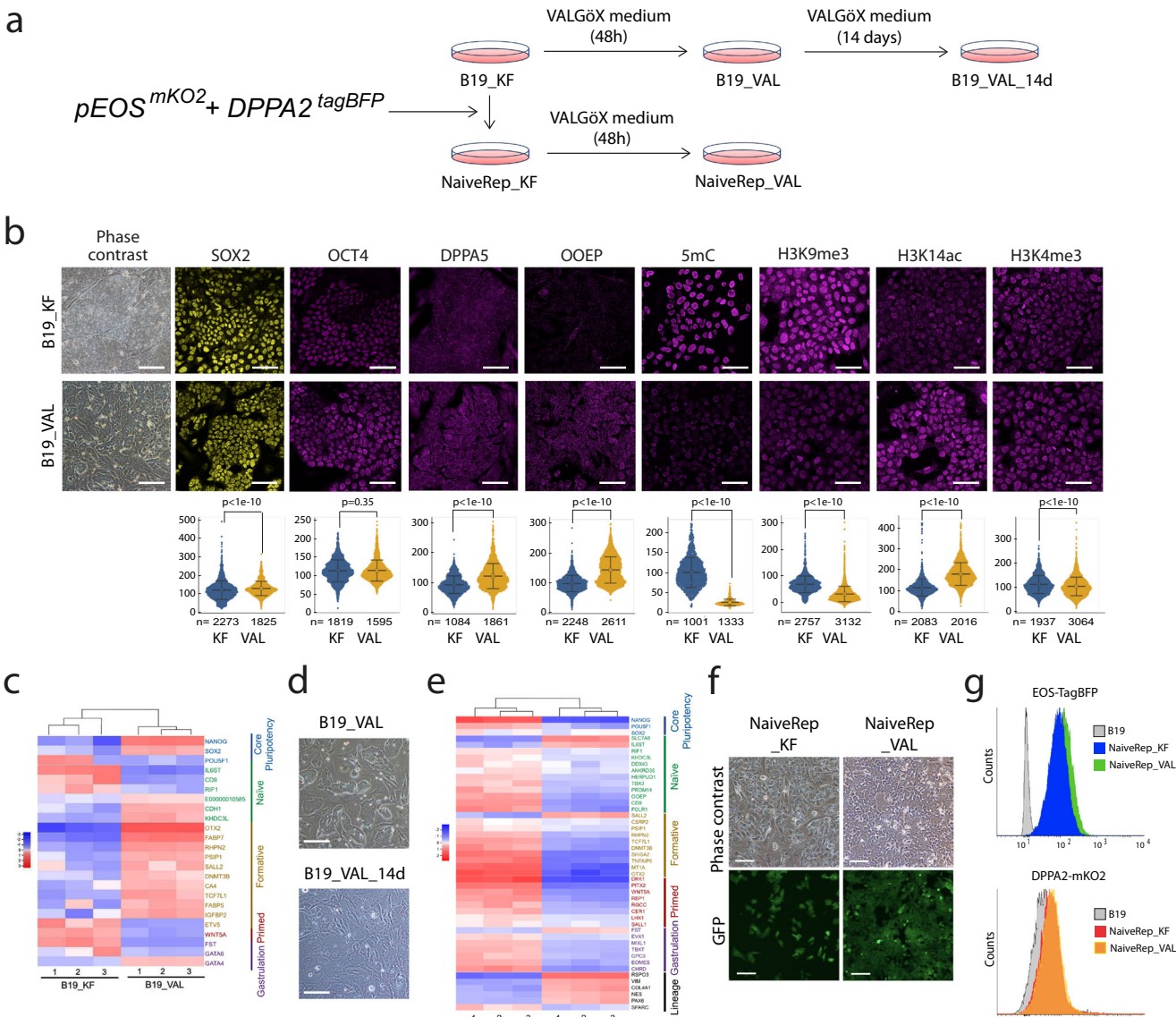

**Fig. 1 | A culture regimen and reporter cell line for naïve pluripotency in rabbit. a** Experimental scheme of B19_VAL, B19_VAL_14d, NaiveRep_KF, and NaiveRep_VAL cell line generation. **b** Immunostaining for SOX2, OCT4, DPPA5, OOEP, 5mC, H3K9me3, H3K14ac, and H3K4me3 in B19 cells before (_KF) and after (_VAL) switching to VALGöX culture medium. Scale bars: 50 μm for immunostainings, 100 μm for phase contrast. Violin plots show fluorescence intensity distribution. Means and standard deviations were calculated from measurements in at least 1,000 cells per condition (exact *n* indicated for each, from independent experiments: 3 for OCT4, OOEP, DPPA5, and 5mC; 4 for SOX2; 5 for H3K14ac and H3K4me3; 6 for H3K9me3). Comparisons between KF and VAL conditions were made using a two-sided Welch's *t*-test for unequal variances. No significant differences were observed for SOX2 and H3K4me3 (difference between means <10% and OCT4 (*p* = 0.35). Source data are provided in the Source Data file. **c** Heatmap of differentially expressed genes between B19_KF and B19_VAL cells (three biological replicates). **d** Phase-contrast images of B19 cells 48 h (_VAL) and 14 days (_VAL_14d) after transfer to VALGöX culture medium (5 experiments). Scale bars: 200 μm. **e** Heatmap of differentially expressed genes between B19_VAL and B19_VAL_14d cells. **f** Phase-contrast and epifluorescence images of NaiveRep_KF and NaiveRep_VAL cells (5 experiments). Scale bars: 100 μm. **g** Flow cytometry analysis of parental B19_KF, NaiveRep_KF, and NaiveRep_VAL showing blue and red fluorescence corresponding to EOS-tagBFP and DDPA2-mKO2, respectively. Basal tagBFP expression in NaiveRep_KF cells reflects endogenous ETn promoter activity downstream of the naïve-state-specific distal enhancer of OCT4/POU5F1 in the EOS vector[26].

fluorescence revealed significant variations in the percentage of positive cells, ranging from 1 to 80% (tagBFP) and from 1 to 7% (mKO2) (Fig. 2b). Analysis of the genomic DNA of the 27 clones by PCR revealed an average of 9.3 ± 3.6 different provirus integrations per clone (Fig. 2c). The most frequently observed proviral DNAs included *ESRRG* (24 clones), *ERAS* (23 clones), *KLF2* (22 clones), *BMI1* (18 clones), *PRMT6* (17 clones), *NANOG* (16 clones), *ESRRB* (16 clones), and *AgT* (15 clones). Out of 63 possible combinations of three different proviruses among the eight most frequent, seven were observed in more than 50% of the 27 clones analyzed: *KLF2/ERAS/ESRRB KLF2/ERAS/BMI1*; *ERAS/ESRRG/PRMT6*; *ERAS/ESRRB/*

*ESRRG*; *ERAS/ESRRG/BMI*; *KLF2/ERAS/PRMT6*; and *KLF2/ERAS/ESRRG*. Notably, among the six clones with more than 30% of cells expressing tagBFP, four clones harbored cDNAs for *KLF2, ERAS, ESRRG,* and *PRMT6*; one clone harbored *KLF2, ERAS,* and *ESRRG*; and one clone harbored *ERAS* and *ESRRG*. In summary, the screening identified a combination of four proviruses (*KLF2, ERAS, ESRRG,* and *PRMT6*) that most frequently occurred among EOS-tagBFP-induced clones.

In the second screening experiment, 2.4 × 10⁵ NaiveRep cells were transduced with an updated cDNA library containing 11 additional human cDNAs (*SMAD7, DPPA2, DPPA4, DPPA5, KLF17, TBX3, DAX1, TRIM8, USP21, TFAP2C,* and *OOEP*). The MOI was reduced to 15 viruses

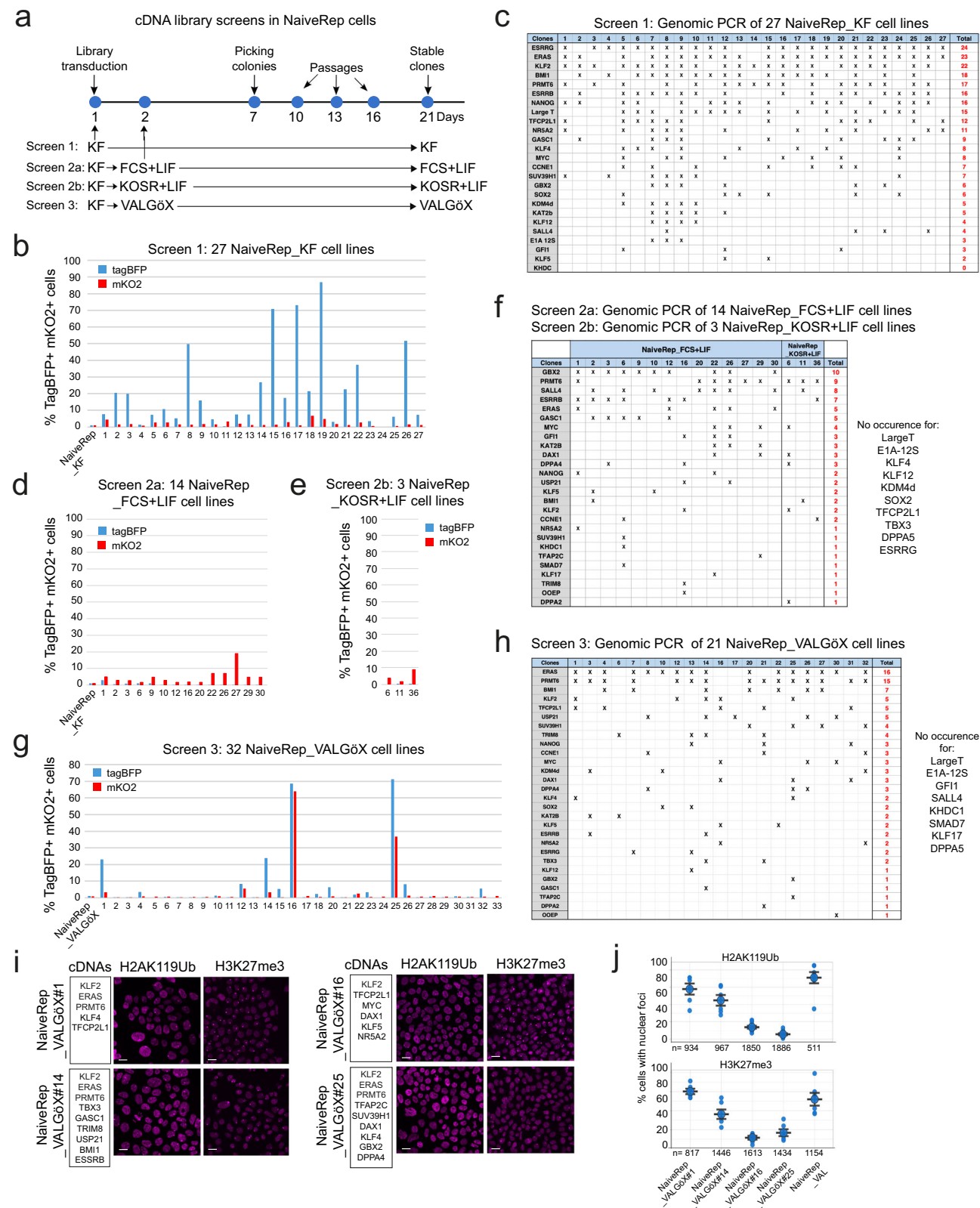

**Fig. 2** (panels a–j)

per cell (-0.4 virus per cDNA per cell) to decrease the average number of cDNAs per clone, thus increasing screening stringency. The transduced cell population was shifted from FGF2/KOSR to either fetal calf serum (FCS) + LIF (screen 2a) or KOSR + LIF (screen 2b) (Fig. 2a). In these two culture conditions, unmodified NaiveRep cells were unable to self-renew and differentiated. Approximately 5000 colonies were examined under epifluorescence, with 14 colonies (screen 2a; Fig. 2d) and three colonies (screen 2b; Fig. 2e) exhibiting tagBFP- and/or

mKO2-associated fluorescence. These colonies were selected and amplified. The average number of proviral integrations was 4.8 ± 2.4. The most frequently observed proviral integrations were *GBX2*, *PRMT6*, *SALL4*, and *ESRRB* (Fig. 2f). No specific combination of proviral integrations could be identified in screens 2a and 2b.

In the third screening experiment, $2.4 \times 10^5$ NaiveRep cells were transduced with the extended cDNA library using a MOI of 15. After 24 h, the cells were transferred from FGF2/KOSR to VALGöX culture

**Fig. 2 | cDNA library screening. a** Experimental scheme depicting the four screening conditions. **b** Flow cytometry analysis of EOS-tagBFP and DPPA2-mKO2 expression in 27 NaiveRep_KF cell lines. **c** Genomic PCR analysis showing proviral integrations in the 27 NaiveRep_KF cell lines. **d** Flow cytometry analysis of EOS-tagBFP and DPPA2-mKO2 expression in 14 NaiveRep_FCS + LIF cell lines. **e** Flow cytometry analysis of EOS-tagBFP and DPPA2-mKO2 expression in 3 NaiveRep_KOSR + LIF cell lines. **f** Genomic PCR analysis identifying the cDNAs integrated in the 14 NaiveRep_FCS + LIF cell lines and 3 NaiveRep_KOSR + LIF cell lines. **g** Flow cytometry analysis of EOS-tagBFP and DPPA2-mKO2 expression in 32 NaiveRep_VALGöX cell lines. **h** Genomic PCR analysis identifying cDNAs in 21 of the 32 NaiveRep_VALGöX cell lines. **i** Immunostaining for H2AK119Ub and H3K27me3 in four NaiveRep_VALGöX cell lines (from two independent experiments). Scale bars: 20 μm. **j** Histograms showing the percentage of cells with H2AK119Ub and H3K27me3 nuclear foci in four NaiveRep_VALGöX cell lines and in NaiveRep_VAL control cells. All cells were quantified individually (exact *n* indicated) from two independent experiments, except for H2AK119Ub analysis in lines #16 and # 25, which were analyzed in three independent experiments. Each dot represents the mean percentage of cells with nuclear foci per replicate; error bars indicate mean ± SEM. Statistical comparisons to NaiveRep_VAL controls were performed using a two-sided Welch's *t*-test for unequal variances. For H2AK119Ub, a reduced percentage of cells with nuclear foci was observed in all NaiveRep_VALGöX cell lines (#1, #14, #16, and #25; $p = 0.17$, 0.01, 0.00006, and 0.00003, respectively). For H3K27me3, Line #1 was not significantly different from controls ($p = 0.30$), while lines #14, #16, and #25 showed reduced percentage of cells with nuclear foci ($p = 0.09$, 0.001, and 0.002, respectively). Source data are provided in the Source Data file.

medium and cultured for an additional 7 days (Fig. 2a). Almost 9000 colonies were examined under epifluorescence, and 32 displayed strong tagBFP- and mKO2-associated fluorescence. These colonies were selected and amplified for further analysis, later termed NaiveRep_VALGöX clones. In these 32 cell clones, the percentage of fluorescent cells ranged from 1 to 71% (tagBFP) and from 1 to 64% (mKO2) (Fig. 2g). Of these, 21 were analyzed for proviral integrations, with an average of 4.8 ± 2.0 integrations per clone. The most frequently observed proviral integrations were *ERAS* and *PRMT6* (Fig. 2h). Notably, all clones exhibiting high EOS^tagBFP and DPPA2^mKO2 expression contained the *KLF2* provirus, and all but one also harbored the *ERAS* and *PRMT6* proviruses. Consequently, this screen identified a combination of three proviruses (*KLF2*, *ERAS*, and *PRMT6*) in clones exhibiting higher expression of both EOS^tagBFP and DPPA2^mKO2 naïve markers.

Four cell lines that displayed activation of both fluorescent reporters in the VALGöX culture condition (NaiveRep_VALGöX#1, NaiveRep_VALGöX#14, NaiveRep_VALGöX #16, and NaiveRep_VALGöX#25; Supplementary Fig. S2e) were selected for further analysis. All four cell lines exhibited consistent growth for at least six passages (Supplementary Fig. S2f). Immunofluorescence analysis of histone marks H3K27me3 and H2AK119Ub showed a reduction in the percentage of cells with nuclear foci, suggesting the reactivation of the second X chromosome in some cells (Fig. 2i, j). Notably, a negative correlation was observed between the percentage of EOS^tagBFP-positive cells and the percentage of cells harboring a nuclear foci H3K27me3 and H2AK119Ub (Supplementary Fig. S2g), further supporting the naïve-like epigenetic characteristics of the fluorescent cell population.

## KLF2, ERAS, and PRMT6 stabilize iPSCs in a naïve-like state

To investigate the role of KLF2 (K), ERAS (E), and PRMT6 (P) individually or in combinations in restoring naïve pluripotency characteristics in rabbit iPSCs, B19 were transfected with plasmids expressing V5-tagged KLF2 and neoR (*pKLF2:V5-neo*), HA-tagged ERAS and hygroR (*pHA:ERAS-hygro*), and Flag-tagged PRMT6 and puroR (*pFlag:PRMT6-puro*), respectively. Following selection in VALGöX medium, resistant colonies were pooled and propagated for ten passages before phenotypic analysis (Supplementary Fig. S3a). Similar to control B19_VAL cells, B19_VAL cells expressing *HA:ERAS* and *Flag:PRMT6* transgenes individually or in combination (referred to as E, P, and EP, respectively), gradually differentiated between passage 5 (P5) and P10 (Fig. 3a and Supplementary Fig. S3b). In contrast, B19_VAL cells expressing *KLF2:V5* alone or with *HA:ERAS*, and *Flag:PRMT6* (K, KE, KP, and KEP, respectively) maintained self-renewal without visible differentiation, highlighting KLF's crucial role in long-term self-renewal in VALGöX. Interestingly, cells expressing the three transgenes (KEP) showed higher levels of the naïve pluripotency cell surface marker CD75[42,43] compared to K, KE, and KP configurations (Fig. 3a, b). Furthermore, KEP cells exhibited fewer cells with nuclear foci after immunostaining for H3K27me3 and H2AK119Ub, suggesting a higher rate of X chromosome reactivation when *KLF2:V5, HA:ERAS*, and *Flag:PRMT6* are co-expressed (Fig. 3a, c). Double immunostaining of CD75 and H3K27me3

revealed a strong correlation between CD75 expression and the reactivation of the second X chromosome (Fig. 3d).

Further investigation into the synergistic effect of the KEP transgenes and the VALGöX culture regimen involved transfecting B19 cells cultured in KF with *pKLF2:V5-neo*, *pHA:ERAS-hygro*, and *pFlag:PRMT6-puro*. Four triple-resistant clones—KEPconstitutive (KEPc)#37_KF, KEPc#39_KF, KEPc#42_KF, KEPc#44_KF—were isolated and subsequently transitioned to VALGöX conditions, resulting in KEPc#37_VAL, KEPc#39_VAL, KEPc#42_VAL, and KEPc#44_VAL cells, respectively (Fig. 3e). Immunofluorescence analysis revealed membrane-bound expression of HA:ERAS and nuclear expression of KLF2:V5 and Flag:PRMT6 in both KEPc_KF and KEPc_VAL cells (Fig. 3f and Supplementary Fig. S3c). Additionally, H3R2me3 mark levels were significantly increased, reflecting the activity of PRMT6's arginine methyltransferase protein. AKT phosphorylation also significantly increased, reflecting ERAS activity (Supplementary Fig. S3d)[44], whereas expression of ESRRB, a target gene KLF2, was higher in KEPc cells than in control cells (Supplementary Fig. S3f). Transitioning from KF to VAL substantially reduced the ratio of cells with nuclear foci observed after immunostaining for H3K27me3 and H2AK119Ub, suggesting reactivation of the second X chromosome in the majority of KEPc_VAL cells (Fig. 3f, g and Supplementary Fig. S3c). Overall, these findings demonstrate that the expression of KLF2:V5, HA:ERAS, and Flag:PRMT6, combined with the VALGöX culture regimen, promotes epigenetic remodeling consistent with naïve-like pluripotency. We further investigated the synergy between transgenes and VALGöX by examining the importance of each component of the VALGöX culture medium (i.e., MEF-conditioning, Vitamin C, activin A, LIF, Gö6983, and XAV939) in maintaining KEPc#37_VAL cells in the naive state. We observed that removing any one of these molecules led to the decline of pluripotency markers and upregulation of differentiation markers within 3 weeks (Supplementary Fig. S4).

To further validate the transition from primed to naïve-like pluripotency states, we analyzed the transcriptomes of B19_KF, B19_VAL, KEPc_KF, and KEPc_VAL cell lines using RNA sequencing. Principal component analysis (PCA) identified four distinct clusters among these PSC lines. The least significant difference was observed between B19_KF and B19_VAL, which exhibited 597 differentially expressed genes (DEGs), while the most substantial difference was between B19_KF cells and KEPc_VAL cells, with 1707 DEGs (Fig. 3h and Supplementary Fig. S3e). In KEPc_VAL cells, there was an upregulation in the expression of naïve pluripotency genes such as *DPPA2*, *DPPA5*, *PRDM14*, *KHDC3L*, *FOLR1*, *ST6GAL1 (CD75)*, and *TDRKH* compared to both B19_KF and B19_VAL. Conversely, the expression of primed pluripotency and gastrulation genes, including *CER1*, *DKK1*, *TBXT*, and *EVX1* was reduced (Fig. 3i and Supplementary Fig. S3f). Notably, genes like *DPPA5*, *PRDM14*, and *KHDC3L*, showed a pronounced increase in expression in KEPc_VAL cells compared to all other cell types, including KEPc_KF, underscoring the synergistic effect of KEP transgenes and the VALGöX culture regimen.

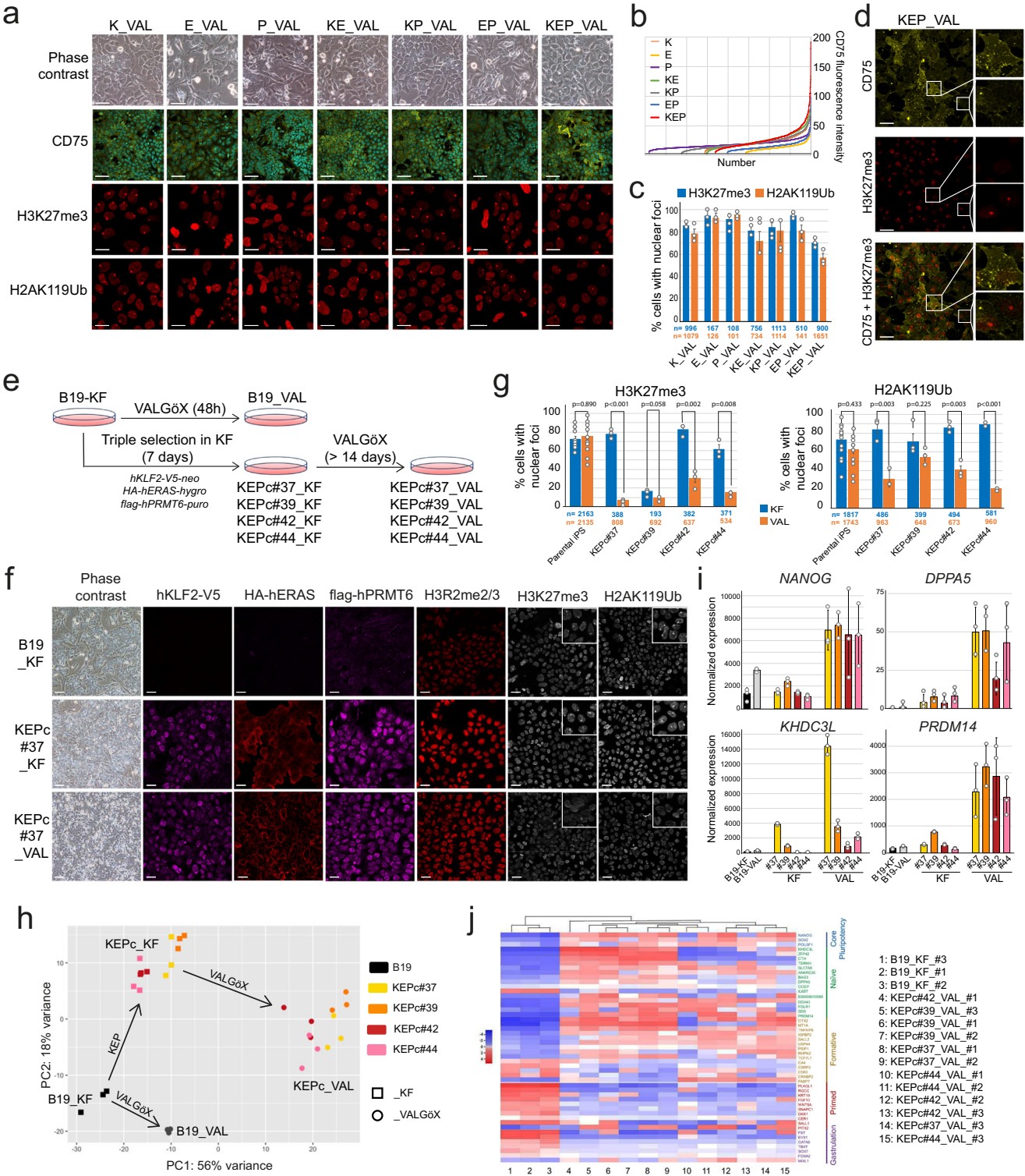

Furthermore, a heatmap of DEGs between B19_KF and KEPc_VAL was generated based on a list of 80 genes differentially expressed between various developmental stages of rabbit embryos—inner cell mass (E3.5), early epiblast (E4.0), mid-epiblast (E5.0), and late epiblast (E6.0 and E6.6) as identified in prior studies[24] (Fig. 3j). Fifty of these genes showed differential expression between B19_KF and KEPc_VAL cells. Notably, genes predominantly overexpressed in B19_KF cells aligned with stages E6.0 and E6.6, termed "primed" and "gastrulation," while the majority of genes overexpressed in KEPc_VAL cells aligned with stages E3.5 and E4.0, termed "formative" and "naïve." Overall, these results demonstrate that the expression of *KLF2:V5, HA:ERAS,*

and *Flag:PRMT6* in rbiPSCs cultured in VALGöX effectively facilitates their conversion from primed to formative/naïve-like pluripotency.

## KEP cells exhibit enhanced embryo colonization ability

We next investigated the ability of KEPc cells to colonize rabbit embryos. KEPc#37_KF, KEPc#37_VAL, KEPc#44_KF, and KEPc#44_VAL cells were injected into early morula-stage embryos at a ratio of eight cells per embryo. The injected embryos were then cultured for three days and developed into late blastocysts (3 days in vitro, DIV) (Fig. 4a and Supplementary Fig. S5). In three independent experiments, 63% of the blastocysts displayed GFP⁺ cells following the injection of

**Fig. 3 | Stabilization of a naïve-like pluripotent state with a KLF2/ERAS/PRMT6 gene cocktail. a** Phase-contrast and confocal images of B19_VAL before and after single, double, and triple transfection with PiggyBac plasmids expressing KLF2-V5 (K), HA-ERAS (E), and flag-PRMT6 (P). Immunostainings for CD75, H3K27me3 and H2AK119Ub is shown. Scale bars: phase contrast, 100 μm; CD75, 50 μm; histone modifications, 20 μm. **b** Graphical representing the distribution of fluorescence intensities (between 0 and 200 arbitrary units) for CD75 immunostaining. Source data are provided in the Source Data file. **c** Percentage of cells with nuclear foci of H3K27me3 and H2AK119Ub across control, single-, double-, and triple-transfected cell populations. Data are from three independent replicates, with each dot representing one replicate. "*n*" indicates the number of cells analyzed per condition. Bars and error bars represent mean ± SEM. A two-sided Welch's *t*-test revealed significantly lower percentages of cells with nuclear foci in KEP_VAL compared to all other conditions (*p* ≤ 0.01). Source data are provided in the Source Data file. **d** Immunostaining of KEP_VAL cells for CD75 and H3K27me3. Scale bar: 40 μm. **e** Experimental design for generating KEPc_KF and KEPc_VAL stable cell lines. **f** Phase-contrast and confocal images of control B19_KF, KEPc#37_KF, and

KEPc#37_VAL cells stained for KLF2-V5, HA-ERAS, flag-PRMT6, H3R2me2/3, H3K27me3, and H2AK119Ub. Scale bars: phase contrast, 100 μm; immunostainings, 30 μm. **g** Quantification of cells with H3K27me3 and H2AK119Ub nuclear foci in B19 parental iPSCs, KEPc_KF, and KEPc_VAL cell lines. Values represent means ± SEM from at least 300 cells per condition (exact *n* provided), based on three independent replicates (additional replicates for parental B19 cells). Each dot indicates a replicate. Two-sided Welch's *t*-test was used to compare KF and VAL conditions (*p* values indicated). Source data are provided in the Source Data file. **h** Two-dimensional PCA of transcriptomic profiles from five cell populations (B19, KEPc#37, #39, #42, and #44), cultured in two conditions (KF: square; VALGöX: circle), with three biological replicates each. **i** Histograms showing gene expression levels (RNA-seq) in B19_KF, B19_VAL, KEPc_KF, and KEPc_VAL lines. Mean ± SD are from three independent replicates. **j** Heatmap of differentially expressed genes among B19_KF and four KEPc_VAL lines (#37, #39, #42, and #44). Genes were selected from a set of 80 differentially expressed genes identified during embryonic development (inner cell mass at E3.5, early epiblast at E4.0, mid-epiblast at E5.0, and late epiblast at E6.0-E6.6[24]).

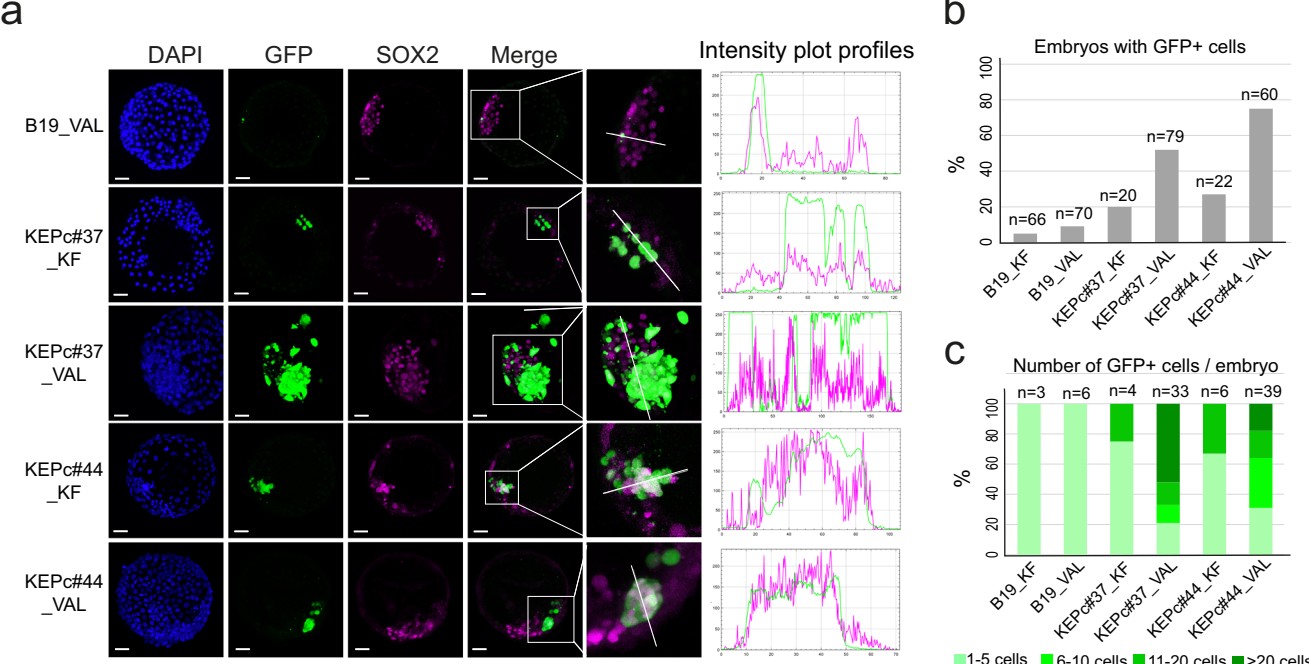

**Fig. 4 | Colonization of rabbit embryos by iPSCs overexpressing KLF2, ERAS and PRMT6. a** Confocal images of late-blastocyst-stage rabbit embryos (E5.0, 3DIV) following microinjection at the early morula-stage (E2.8) with B19_KF, KEPc#37_KF, KEPc#37_VAL, KEPc#44_KF, and KEPc#44_VAL cells. Scale bars: 50 μM. On the right, intensity profile plots were generated along a line drawn through the inner cell mass (ICM) of each embryo using Fiji. Fluorescence signals across channels (green: GFP; purple: SOX2) indicate co-localization of injected GFP⁺ cells with SOX2⁺

regions, suggesting integration into the pluripotent compartment. Images are representative of three independent experiments per cell line and condition. **b** Percentage of rabbit embryos containing GFP⁺ cells, based on three independent experiments for each cell line and condition. **c** Distribution of GFP⁺ embryos according to the number of GFP+ cells observed, corresponding to the embryos quantified in (**b**).

KEPc_VAL cells, compared to 23% for KEPc_KF cells and only 5% for parental iPS B19_KF cells (Fig. 4b). A significant difference was observed in the number of GFP⁺ cells among the three cell types. The majority of embryos injected with eight parental B19_KF or KEPc_KF cells contained fewer than five GFP⁺ cells after 3DIV (Fig. 4c). Only three embryos (7%) injected with KEPc_KF cells had more than five (but fewer than ten) GFP⁺ cells. In stark contrast, embryos injected with eight KEPc_VAL cells demonstrated notably enhanced colonization. For KEPc#37_VAL, among GFP⁺ embryos, up to 66% of them contained over 10 GFP⁺ cells, and 50% had more than 20 GFP⁺ cells. The vast majority of the GFP⁺ cells expressed SOX2 (Fig. 4a and Supplementary Fig. S5). Overall, these results indicate that the KEP transgenes, when combined

with the VALGöX culture regimen, significantly enhance both the growth and embryo colonization capabilities of rbiPSCs.

## A reversible KEP expression system for in vivo studies

We designed rbiPSCs to express *KLF2*, *ERAS* and *PRMT6* transgenes in an inducible manner. To achieve this, B19-GFP_KF cells were transfected with three plasmids encoding Shield1-activatable DD:KLF2:V5 (*pDD:KLF2:V5-neo*), DD:HA:ERAS (*pDD:HA:ERAS-hygro*), and DD:Flag:PRMT6 (*pDD:Flag:PRMT6-puro*). These vectors included a mutated FKBP12-derived destabilization domain (DD) fused to the coding sequence of KLF2:V5, HA:ERAS, and flag:PRMT6, enabling interaction and stabilization by Shield1[45]. After transfection of B19_KF

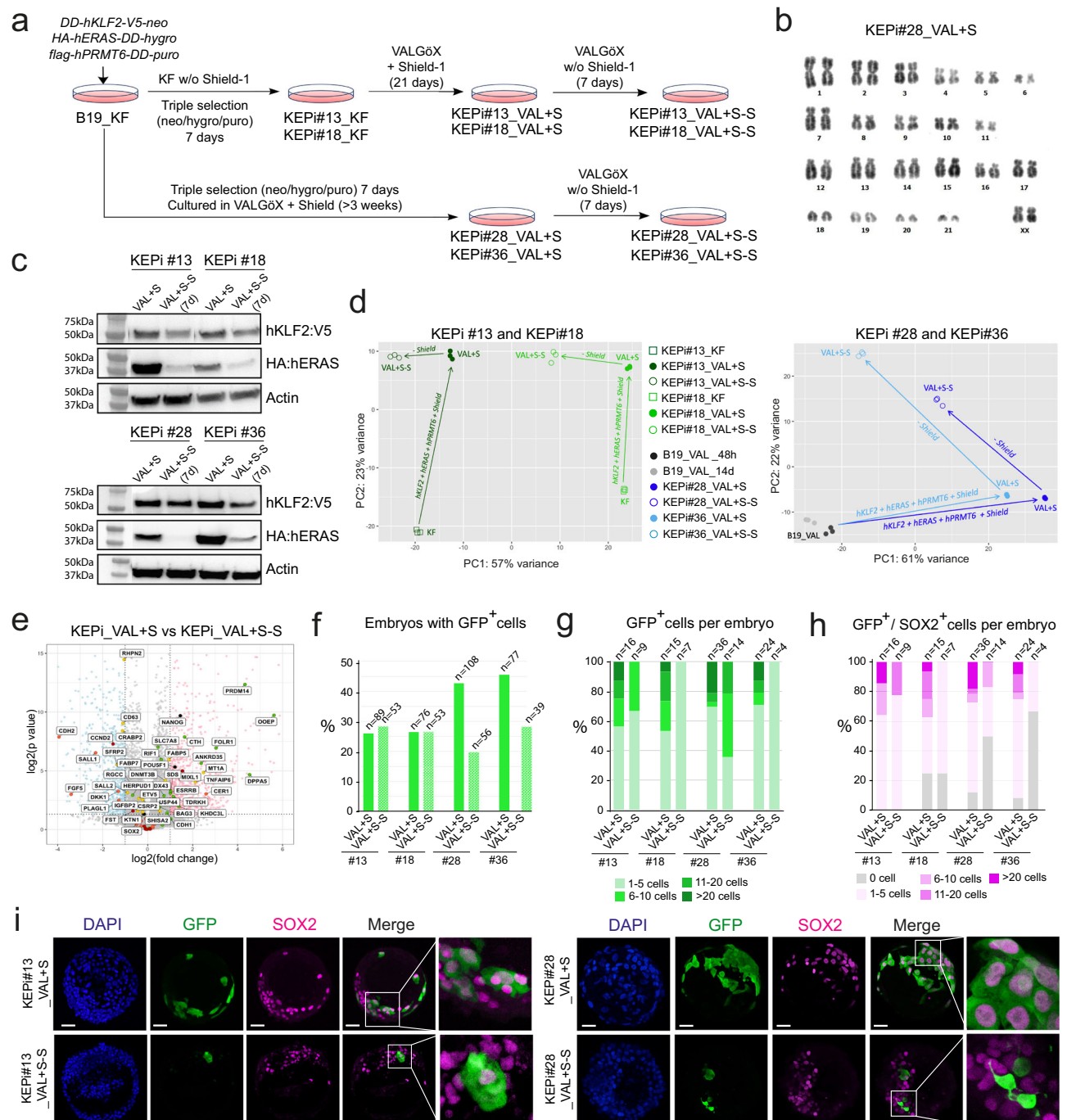

**Fig. 5 | Design of a reversible KEP expression system for in vivo studies.**
**a** Experimental scheme illustrating the generation of KEPi_VAL + S and
KEPi_VAL + S-S cell line. **b** G-banding karyotype analysis of KEPi#28_VAL + S cells.
**c** Western blot analysis of KLF2-V5 and HA-ERAS expression in KEPi#13, KEPi#18,
KEPi#28, and KEPi#36 cells, before and after withdrawal of Shield1 (two biological
replicates). Source data are provided in the Source Data file. **d** Two-dimensional
PCA of transcriptomic profiles from the indicated cell lines (three biological
replicates). **e** A volcano plot of differentially expressed genes (DEGs) between
KEPi_VAL + S and KEPi_VAL + S–S cells, based on a Mann–Whitney U-test.
**f** Percentage of rabbit embryos containing GFP+ cells after microinjection of
each cell line (three independent experiments). **g** Distribution of embryos
according to the number of GFP+ cells per embryo (three independent experi-
ments). **h** Distribution of embryos according to the number of GFP+/SOX2+ cells
per embryo (three independent experiments). **i** Confocal images of late-blastocyst-
stage rabbit embryos (E5.0, 3DIV) after microinjection of KEPi_VAL + S and
KEPi_VAL + S-S cells into morula-stage (E2.8) embryos. Scale bars: 50 μm.
Images are representative of three independent experiments per cell line and
condition.

cells in the absence of Shield1, two triple-resistant cell lines,
KEPi#13_KF and KEPi#18_KF, were established. These cell lines were
then transferred to VALGöX medium supplemented with Shield1 and
cultured for three weeks, producing KEPi#13_VAL + S and KEPi#18_-
VAL + S (Fig. 5a). In a subsequent experiment, we transfected B19_KF
cells with the same plasmids and immediately cultured them in VAL-
GöX medium with Shield1 for at least three weeks, resulting in

KEPi#28_VAL + S and KEPi#36_VAL + S cells. A karyotype analysis of
KEPi#28_VAL + S cells demonstrated stable chromosomal count and
structure (Fig. 5b). We then cultured these cells in Shield1-free medium
for 7 days, resulting in KEPi#13_VAL + S-S, KEPi#18_VAL + S-S,
KEPi#28_VAL + S-S, and KEPi#36_VAL + S-S, respectively. Following
Shield1 withdrawal, there was almost complete degradation of ERAS
and PRMT6 and partial degradation of KLF2 (Fig. 5c and

Supplementary Fig. S6a). The levels of H3R2me2/3 histone mark and phosphorylated AKT decreased (Supplementary Fig. S6a, c, d). However, despite the withdrawal of Shield1, the number of cells with nuclei foci remained unchanged, suggesting that the XaXa status of KEPi cells was not altered (Supplementary Fig. S6b). Transcriptome analysis of KEPi#_KF, KEPi#_VAL + S, and KEPi#_VAL + S-S cell lines was visualized using two-dimensional PCA (Fig. 5d). Activation of the transgenes in VALGöX culture conditions caused significant transcriptome changes in the four analyzed clones. Changes induced by Shield1 withdrawal were minor in KEPi#13 cells, moderate in KEPi1#18 cells, and substantial in KEP#28 and KEPi#36 cells. Naïve pluripotency markers such as *DPPA2*, *DPPA5*, *KHDC3L*, *KLF4*, *PRDM14*, and *ZFP42* were downregulated, while primed pluripotency markers including *SALL1*, *CDH2*, *FGF5*, *PLAGL1*, and *DKK1* were upregulated in KEPi#_VAL + S-S cells compared to KEPi#_VAL + S (Fig. 5e and Supplementary Fig. S6e). This indicates that degradation of the transgene product destabilized KEPi_VAL cell renewal in the naïve state of pluripotency but did not lead to visible differentiation for the duration of the experiment.

To assess the embryonic colonization capacity of KEPi_VAL + S and KEPi_VAL + S-S cells, we injected eight cells from each cell line into rabbit morulae. A total of 35% of the blastocysts (*n* = 350) exhibited GFP⁺ cells following injection of KEPi_VAL + S cells, with specific rates of 25.8, 26.3, 42.6, and 45.5 in clones #13, #18, #28, and #36, respectively. In comparison, 25% of blastocysts (*n* = 201) displayed GFP⁺ cells after injection of KEPi_VAL + S-S, with rates of 28.3, 26.4, 19.6, and 28.2% in the same respective clones (Fig. 5f–i and Supplementary Fig. S7). The GFP⁺ cells were either organized in clusters or scattered throughout the embryo. The average number of GFP⁺ cells per embryo was 6.5 (ranging from 1 to 27) following injection of KEPi_VAL + S cells (Fig. 5g). The vast majority of these GFP⁺ cells expressed SOX2, averaging 5.8 cells per embryo, with counts ranging from 1 to 27, and were predominantly localized in the epiblast (Fig. 5h). The average number of GFP⁺ and GFP⁺/SOX2⁺ cells slightly decreased in embryos injected with KEPi_VAL + S-S cells, showing 4.7 GFP⁺ cells per embryo (ranging from 1 to 10). However, these counts remained substantially higher than those observed in embryos injected with parental B19_VAL cells, which averaged 1.25 GFP⁺ per embryo (ranging from 1 to 2) and one GFP⁺/SOX2⁺ cell per embryo (ranging from 0 to 2). In conclusion, we have developed a reversible KEP expression system that is instrumental in investigating the fate of KEP_VAL cells in chimeric embryos.

## High-CD75 KEP cells show very high chimeric competence

We previously observed that a subpopulation of B19_VAL_KEP cells exhibited high levels of CD75 (Fig. 3b–d). Motivated by this observation, we investigated the presence and embryo colonization capabilities of a CD75high cell subpopulation within KEPi_VAL + S cells. Following CD75 analysis by flow cytometry, both KEPi#13_VAL + S and KEPi#28_VAL + S cells formed a continuum. The top 2.1% of KEPi#13_VAL + S and top 3.5% of KEPi#28_VAL + S populations were FACS-sorted, and then cultured under two conditions: for 24 h (Condition I) and for 5 days (Condition II) before injection into rabbit morulae (Fig. 6a, b and Supplementary Fig. S8). In condition I, 77 and 100% of embryos displayed GFP⁺ cells post-injection of KEPi#13_CD75high and KEPi#28_CD75high cells, respectively (Fig. 6c, d and Supplementary Fig. S9a). Notably, 70% of embryos from KEPi#13 and all from KEPi#28 had incorporated more than 20 GFP⁺ cells (Fig. 6e). In Condition II, 51 and 100% of embryos showed GFP⁺ cells post-injection of KEPi#13_CD75high and KEPi#28_CD75high cells, respectively, with 32 and 63% of these embryos incorporating over 20 GFP⁺ cells. These results demonstrate that CD75high cells in KEPi_VAL + S cell populations possess significantly enhanced capacities for epiblast colonization.

To explore the potential of CD75high cells in contributing to the three germ layers and organogenesis, we injected 248 morulae with KEPi#28_CD75high cells and cultured them for 24 h. Subsequently, we transferred 224 of these to 14 surrogate female rabbits, resulting in 38

fetuses (Table 1). Note that the recovery rate (17%) is similar to that observed with unmanipulated embryos[46,47]. Out of 38 fetuses collected at E10.5, five exhibited extensive colonization by GFP⁺ cells (Table 1, Fig. 6f, and Supplementary Fig. S9b, c). Moreover, of the 18 fetuses analyzed by genomic PCR, 15 (83%) contained GFP DNA (Table 1 and Supplementary Fig. S9d). Phenotypic analysis of GFP⁺ cells in sections showed differentiation into various cell types, including SOX2⁺ neuronal progenitors, TUJ1⁺ neurons, SMA⁺ and DESMIN⁺ cardiac cells, PAX3⁺ somitic mesoderm, SOX17⁺ and CD31⁺ gut epithelium, and LAMININ⁺ surface ectoderm (Fig. 6g and Supplementary Fig. S9b, c). These findings underscore the high efficiency of KEPi_CD75high cells in contributing to tissue development of tissues from ectodermal, mesodermal, and endodermal origins in rabbit fetuses.

To investigate the ability of CD75high cells to contribute to produce viable chimeras, 273 morulae were injected with KEPi#28_CD75high cells, and 220 of these were subsequently transferred to 14 surrogate female rabbits. Of the 13 newborn rabbits obtained, two died shortly after birth due to developmental abnormalities in the heart and intestine. Five were abandoned by their mothers and had to be sacrificed—one on the 1st day and the others 3 days after birth. The remaining six developed into viable young rabbits (Table 1). Both the two rabbits that died naturally and the five that were sacrificed were confirmed as chimeric by genomic PCR, performed on total DNA extracted from various organs including the brain, ears, lungs, heart, trachea, intestine, stomach, kidneys, spleen, liver, muscle, pancreas, testes/ovaries bladder, tongue, skull skin, tail, and eyes (Supplementary Fig. S10 and Supplementary Table 1). Additionally, a variable proportion of GFP⁺ cells was observed in whole mounts (Supplementary Fig. S11a, b) and in sections of the skin, heart, kidneys, brain, liver, and muscles, confirming the PCR data (Supplementary Fig. S11c). These findings demonstrate the ability of KEPi_CD75high cells to generate viable newborn chimeras.

## High-CD75 KEP cells generate germline chimeras

The six viable rabbits, comprising three males and three females, were reared for 7 weeks before undergoing blood sampling, which confirmed that all were chimeric (Fig. 7a). Chimerism was further verified in five of the six young rabbits using buccal swab (Fig. 7b). At 6 months, all six chimeras were euthanized, and their organs were analyzed through quantitative genomic PCR and immunostaining to assess chimerism rates across various tissues. Quantitative PCR revealed that virtually all analyzed organs were positive for GFP, with chimerism rates ranging from 0.01 to 100% (Supplementary Fig. S12a, b and Supplementary Table 2). Notably, the degree of chimerism differed among organs and between individuals. For example, the heart exhibited chimerism rates between 0.01 and 99.8%, the muscles ranged from 0.01 to 88.7%, and the skin from 0.01 to 100%. These findings were further supported by immunofluorescence analysis of tissue sections from muscles, lungs, liver, skin, tongue, and ovaries (Fig. 7c and Supplementary Fig. S12c). Discrepancies between chimerism rates measured by genomic PCR and immunofluorescence analysis were observed in the skin (Supplementary Table 2 and Fig. 7c). These discrepancies can be attributed to the variations in the degree of chimerism across different parts of the body. Notably, numerous GFP⁺ oogonial cells were identified within the follicles of chimera #A2 (36 GFP⁺ oogonial cells out of 40 examined) and chimera #A6 (43 GFP⁺ oogonial cells out of 47 examined) (Fig. 7c, d), underscoring the significant contribution of iPSCs to the female germline.

Before euthanasia, the three chimeric females were inseminated with sperm from wild-type males. On day 14 of gestation, 27 embryos were recovered (14 from female #A2 and 13 #A6). Genomic PCR analysis revealed that all embryos were GFP-positive (Fig. 7e), and all the embryos examined displayed GFP fluorescence (Fig. 7f). Notably, there were no apparent differences in external anatomy or developmental stage between the E14 GFP fetuses and the E14

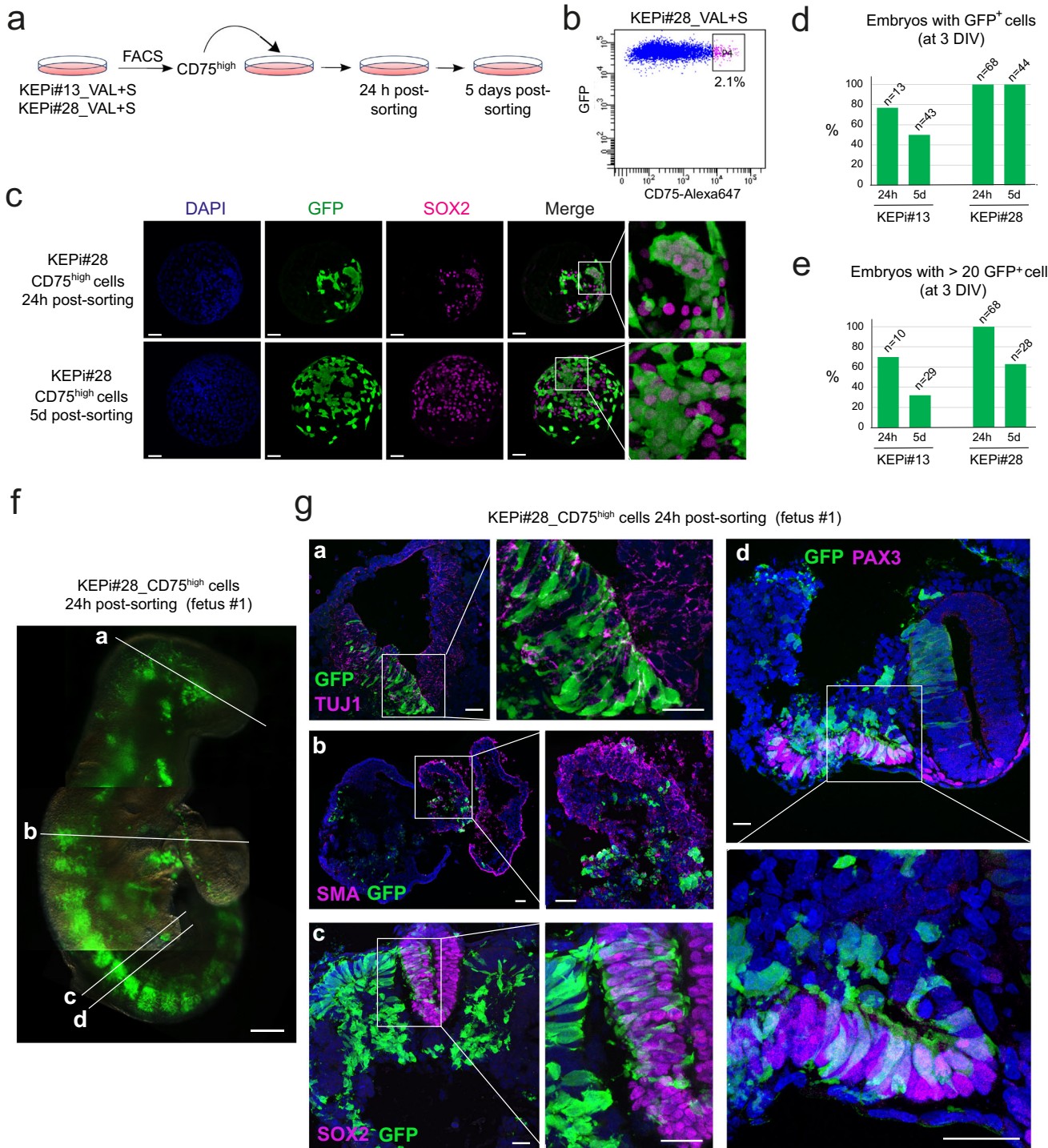

**Fig. 6 | Chimerism induced by KEP_VAL_CD75high cells in rabbit fetuses.**
**a** Experimental scheme for the isolation and analysis of CD75high subpopulations from KEPi#28_VAL + S cells. **b** Flow cytometry plot showing sorting of the CD75high subpopulation from KEP#28_VAL + S cells. **c** Confocal images of late-blastocyst-stage rabbit embryos (E5.0, 3DIV) after microinjection of KEPi#28_ CD75high cells into morula-stage (E2.8) embryos. Images are representative of three independent experiments. Scale bars: 50 μm. **d** Percentage of embryos containing GFP+ cells for each condition. **e** Percentage of embryos containing more than 20 GFP+ cells per embryo. **f** GFP+ chimeric rabbit fetus was recovered at embryonic day 10.5 (E10.5) from two independent experiments. Scale bar: 200 μm. **g** Confocal images of transverse sections from an E10.5 chimeric fetus showing GFP+ cells labeled with anti-GFP antibody in multiple tissues: the neural tube (co-labeled with TUJ1 and SOX2), the heart (co-labeled with SMA), and the somites (co-labeled with PAX3). Images represent three technical replicates. Scale bars: 20 μm.

controls–derived from a wild-type female inseminated with wild-type sperm. This high rate of GFP-positive fetuses is attributed to multiple copies of proviral DNA in KEPi#28 cells, a result of multiple trans-ductions by GAE-GFP lentiviruses. We performed ligation-mediated PCR to identify proviral DNA integration sites and detected five distinct sites across the genome: chromosome 1, 6, 10, 11, and 15, respectively. All five were confirmed by DNA sequencing (Supple-mentary Fig. S13). We subsequently calculated that 98.5% of gametes should carry at least one copy, a figure consistent with our observed transmission rate based on GFP PCR.

**Table 1 | Results of embryo injections with KEPi_VAL_CD75[high] cells and transferred to foster mothers**

| | Embryos injected | Embryos transferred | Fetuses E10.5 | Stillborn | Newborns sacrificed | Newborns alive | Chimeric | | | Sex and #Name | Germline Transmission |
|---|---|---|---|---|---|---|---|---|---|---|---|
| | | | | | | | Fluo | PCR | % | | PCR on F1 |
| Exp #1 | 92 | 80 | 13 | | | | 1 | N.D. | | | |
| Exp #2 | 156 | 144 | 25 | | | | 4 | 15/18 | 83 | | |
| Exp #3 | 60 | 48 | | | | 1 | 1 | 1 | 100 | Male #A1 | - |
| Exp #4 | 60 | 48 | | | | 2 | 1 | 2 | | Female #A2 | Yes |
| | | | | | | | | | | Female #A3 | N.D. |
| Exp #5 | 73 | 60 | | 1 | 1 | | 2 | 2 | | | |
| Exp #6 | 80 | 64 | | 1 | 4 | 3 | 6 | 8 | | Male #A4 | - |
| | | | | | | | | | | Male #A5 | - |
| | | | | | | | | | | Female #A6 | Yes |

*ND* Not Determined

To further confirm germline transmission. we analyzed the GFP provirus integration sites in 11 fetuses from female #A6 (F1#15 to F1#25 in Fig. 7e) using ligation-mediated PCR. Four integration sites were successfully identified in eight fetuses (Fig. 7g): fetuses #16, #18, #19, #20, and #22 showed integration on chromosome 6; fetuses #15, #16, #18, #19, #22, #23, and #24 on chromosome 10; fetuses #18, #19, #20, and #22 on chromosome 11; and fetuses #16, #18, and #20 on chromosome 15. Notably, the integration sites on chromosomes 6, 10, 11, and 15 were validated by DNA sequencing from fetuses #16 and #19. In addition, the fifth integration site–originally identified in KEPi#28 cells was detected in fetuses #16, #18, #19, #20, and #23 by genomic PCR, employing both a GFP primer and a site-specific primer (Fig. 7h). Consequently, the total number of integration sites identified in the F1 fetuses ranged from 0 to 5 among the five sites originally identified in KEP#28 cells (Table 2). Although we did not detect integration sites in three fetuses (#17, #21, and #25), these fetuses were GFP-positive by genomic PCR analysis (Fig. 7e), suggesting that they carry integration sites that eluded detection via our ligation-mediated PCR protocol.

Overall, these results demonstrate the high efficiency of KEPi_CD75[high] cells in generating viable germline chimeras and transmitting their genome to the chimera offspring IS, integration sites.

### Transcriptomic signature of germline-competent KEP cells

To identify potential determinants of enhanced embryonic colonization, we analyzed the transcriptome of B19_KF, B19_VAL, KEPc_KF, and KEPc_VAL cells. With KEPc_VAL cells showing a significantly higher colonization capacity compared to the other cell types, we focused on genes synergistically regulated by both KEP transgenes and VALGöX culture conditions. After removing DEGs associated solely with the "VALGöX effect" (between B19_KF and B19_VAL) and the "KEP effect" (between B19_KF and KEPc_KF) from the 1707 DEGs identified between B19_KF and KEPc_VAL, we isolated 1317 genes representing the "KEP + VALGöX synergistic effect," with 783 upregulated and 534 downregulated (Fig. 8a, b). KEGG pathway enrichment analysis of the upregulated genes highlighted significant involvement in amino acid metabolism pathways, including glycine, serine, threonine, cysteine, methionine, and arginine (e.g., *ACYP1, ALDH7A1, ALDH8A1, AMT, CKMT1B, CNDP1, CTH, GLDC, GNMT, GSS, PSAT1, PYCR2, SDS,* and *SMS*), and NF-kB signaling (e.g., *BTK, EDARADD, ZAP70,* and *LTBR*) (Fig. 8c, d and Supplementary Fig. S14). Conversely, downregulated genes were predominantly linked to the MAPK pathway (e.g., *EGFR, PDGFRB, TGFBR2, ARAF, MAPKAPK2, DUSP2, MAP3K8,* and *ELK4*), both canonical and non-canonical WNT pathways (*e.g., WNT5B, FZD3, FZD4, AXIN2, GPC4, DAAM1,* and *PPP3CA*), the HIPPO pathway (e.g., *LATS2, TAZ, KIBRA, TEAD1, TEAD4,* and *PPP2R2B*), and axon guidance molecules (e.g., *EFNB2, SEMA6D, NTN1, PLXNA2, EPHB1,* and *EPHA4*) (Fig. 8d and Supplementary Fig. S14). This indicates a shift in KEPc_VAL cells toward enhancing survival pathways while reducing pathways associated with differentiation and growth signaling.

We also analyzed B19_VAL and KEPi_VAL + S cells, identifying 339 upregulated and 460 downregulated DEGs. Pathway analysis showed these changes also emphasized altered amino acid metabolism and decreased signaling through MAPK, WNT, HIPPO, NETRIN, and EPHRIN pathways (Fig. 8e, f). Further, by comparing DEGs between KEPi_VAL + S and KEPi_VAL + S-S cells, we observed downregulation of genes involved in amino acid metabolism (e.g., *ALDH7A1* and *CTH*) upon transgene product degradation and upregulation of genes in WNT, MAPK, and HIPPO pathways (e.g., *EGFR, PRICKLE2,* and *LATS2*) (Supplementary Fig. S15a–c). This transcriptomic reversal underscores the pivotal role of these pathways in maintaining the pluripotent state conducive to embryonic colonization. Interestingly, variability in the stability of these transcriptomic changes was observed among the KEPi_VAL cell lines: the average +S/ + S-S fold-change calculated from the 42 DEGs varied significantly between cell lines, with the minimum variation in KEPi#13_VAL and KEPi#18_VAL and the maximum in KEPi#28_VAL and KEPi#36_VAL cells (Supplementary Fig. S15d). These differences in stability correlated with their colonization efficiency, particularly noted in KEPi#28_VAL and KEPi#36_VAL compared to other lines.

In summary, our findings suggest that enhanced embryonic colonization capacity in KEPc_VAL and KEPi_VAL + S cells is linked to higher expression of genes involved in amino acid metabolism and NF-κB signaling, alongside reduced activity in key developmental signaling pathways. This transcriptomic signature might be crucial for the heightened colonization abilities observed.

### Discussion

In this study, we succeeded in producing viable germline chimeras in rabbits using pluripotent stem cells. We have succeeded in this challenge by identifying a combination of three genes—*KLF2, PRMT6,* and *ERAS*—that enable rabbit iPSCs to acquire molecular and functional features of the naïve state, including the capacity to efficiently colonize preimplantation embryos. *Klf2* encodes a transcription factor that participates in maintaining the naïve state of pluripotency in mice[28,48–50]. KLF2 can also promote the conversion from a primed to a naïve state in human PSCs[18,19]. Notably, *KLF2* is highly expressed in rabbit morulae but is downregulated in the ICM and epiblast, which is consistent with its role in the acquisition of naïve pluripotency. *PRMT6* encodes an arginine methyltransferase that influences mESC self-renewal by controlling H3R2me levels. Specifically, knockdown of *Prmt6* leads to the downregulation of pluripotency genes and the induction of expression of differentiation markers[37]. Similar to *KLF2, PRMT6* is highly expressed in rabbit morulae but is downregulated in the ICM and epiblast. Eras, a member of the RAS family, is known to enhance the growth and tumorigenicity of mouse

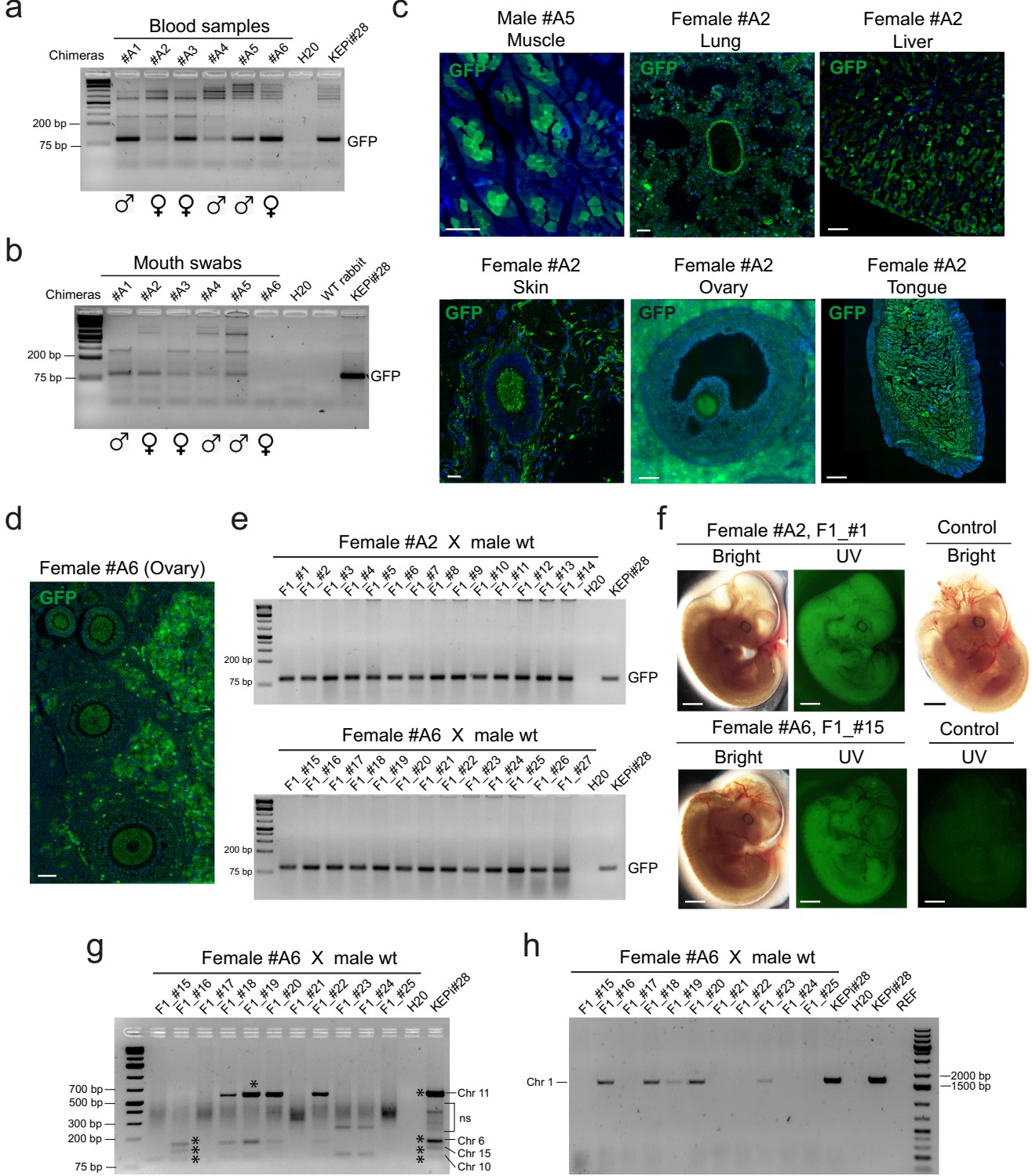

**Fig. 7 | Somatic and germline chimerism induced by KEP_VAL_CD75<sup>high</sup> cells in adult rabbits. a**, **b** Gel electrophoresis of PCR amplification products for GFP DNA sequences in genomic DNA extracted from peripheral blood cells (**a**) and buccal swabs (**b**) of six viable chimeric animals (one biological replicate). **c** Confocal microscopy images of tissue sections from chimeras #A2 and #A5 showing GFP⁺ cells labeled with anti-GFP antibody in muscle (scale bar: 250 μm), lung, liver, ovary (50 μm), skin (25 μm), and tongue (500 μm). Images represent three technical replicates per tissue. **d** Confocal image of a primary ovarian follicle labeled with anti-GFP in a section from female chimera #A6 (scale bar: 75 μm; three technical replicates). **e** Gel electrophoresis of PCR amplification products for GFP DNA sequences from genomic DNA of E14 F1 embryos obtained after insemination of

female chimeras #A2 and #A6 with wild-type sperm (three technical replicates). **f** Epifluorescence images of whole-mount E14 F1 embryos and a non-chimeric control (one biological replicate). **g** Identification of four transgene integration sites in KEPi#28 donor cells and F1 fetuses via ligation-mediated PCR. Asterisks (*) indicate specific PCR bands confirmed by sequencing in KEPi#28 and F1 fetuses #16 and #19. Bands observed in other fetuses were not further analyzed. "ns" denotes non-specific PCR products with unreadable sequences (two technical replicates). **h** PCR amplification of genomic DNA from F1 fetuses confirming the integration of the GFP transgene at a Chromosome 1 locus identified in KEPi#28 cells via ligation-mediated PCR. REF, rabbit embryonic fibroblasts (negative control). Two technical replicates were performed.

**Table 2 | Summary of GFP integration sites in E14 F1 fetuses (#15 to #25) and KEPi#28 cells. IS integration sites**

|  |  | Female #A6 x male wt | | | | | | | | | | | KEPi#28 |
|---|---|---|---|---|---|---|---|---|---|---|---|---|---|
|  |  | #15 | #16 | #17 | #18 | #19 | #20 | #21 | #22 | #23 | #24 | #25 |  |
| PCR | Chr1 | - | + | - | + | + | + | - | - | + | - | - | + |
| LM-PCR | Chr6 |  | + |  | + | + | + |  | + |  |  |  | + |
|  | Chr10 | + | + |  | + | + |  |  | + | + | + |  | + |
|  | Chr11 |  |  |  | + | + | + |  | + |  |  |  | + |
|  | Chr15 |  | + |  | + |  | + |  |  |  |  |  | + |
| Total IS identified |  | 1 | 4 | 0 | 5 | 4 | 4 | 0 | 3 | 2 | 1 | 0 | 5 |

ESCs. However, expression of Eras is not essential to mouse ESC pluripotency[44]. We were unable to detect ERAS transcripts in rabbit embryos. In our study, we showed that overexpression of KLF2 was necessary and sufficient for iPSC self-renewal in VALGöX, but iPSCs underwent a more comprehensive conversion when both PRMT6 and ERAS were included in the reprogramming cocktail. Specifically, overexpression of the three transgenes resulted in the expansion of a cell subpopulation expressing higher levels of the naïve marker CD75 and devoid of H3K27me3 foci, indicating exceptionally immature cells. These immature cells were not observed in KEP_KF cell populations and were barely detectable in KE_VAL, and KP_VAL cells. They emerged when KEP cells were derived in VALGöX culture conditions. Similarly, the embryo colonization ability was virtually zero for parental iPSCs cultivated in KF, low for KEP_KF cells, moderate to high for KEP cells cultivated in VALGöX, and very high for the CD75[high] subpopulation. Overall, these results indicate a synergy between the VALGöX culture regimen and KEP transgenes that enables rabbit iPSCs to enter a functional naive state, including an embryo colonization capability.

It remains unclear how PRMT6 and ERAS, working alongside KLF2, facilitate this conversion. Upregulation of PRMT6 has been linked to global DNA hypomethylation by impairing the chromatin binding of UHRF1, a co-factor of DNMT1, which results in passive DNA demethylation[51]. In various cancer cell lines, including those of breast, lung, colorectal, and cervical cancers, PRMT6 has been found to inhibit the expression of cyclin-dependent kinase inhibitors (CKI) such as p21cip1/waf1, p27kip1, and p18ink4c, or to mitigate the association of p16ink4a with CDK4. This results in accelerated cell cycles and the inhibition of cellular senescence[52–55]. In breast and prostate cancer cell lines, PRMT6 has also been associated with reduced apoptosis and the regulation of motility and invasion[56]. Indeed, the diverse roles of PRMT6 in cell-cycle regulation, migration, and apoptosis appear to be integral to the primed-to-naïve state conversion and the embryo colonization capacity of KEP cells. These functions could potentially enhance the cells' fitness, improve their resistance to harsh environments, and boost their adaptability. The role of ERAS in enhancing the transition from a primed-to-naïve state and fostering subsequent chimeric competence is more elusive. Similar to PRMT6, ERAS is known to activate the AKT pathway and accelerate the mitotic cycle in mouse ESCs[44,57]. We can speculate that its exogenous overexpression could have a substantial impact on the growth of rabbit iPSCs, which might explain why ERAS has consistently emerged in our screens. However, further investigation is necessary to confirm this hypothesis and fully understand the role of ERAS in the context of embryo colonization.

The cells KEPc and KEPi provide a unique opportunity to identify transcriptomic alterations and cellular functions that distinguish cells with high aptitude for embryonic colonization from those with little to no competence. Our findings reveal that chimeric competence is strongly associated with the transcriptional repression of genes involved in MAPK, WNT, HIPPO, and EPH signaling pathways. These

findings corroborate previous studies that emphasized the role of inhibiting the MAPK, WNT, HIPPO pathway in capturing and preserving naïve pluripotency[20,58–60]. The downregulation of EPH receptors and ephrin ligands aligns with the overarching role of EPH-ephrin signaling in counteracting self-renewal in progenitor cells of the nervous system, skin, and intestinal stem cells, as well as in cancer stem cells[61,62]. Further investigation is necessary to clarify the role of EPH-ephrin signaling in naïve pluripotency and embryo colonization capability. Furthermore, the chimeric competence of KEPc and KEPi cells is associated with the transcriptional activation of genes involved in threonine, methionine, glycine and serine metabolism. These findings are consistent with results in mice and humans demonstrating the dependence of ESCs on threonine and methionine metabolism for self-renewal and pluripotency maintenance[63–65]. In mESCs, threonine metabolism supplies methyl groups for maintaining a high ratio of $S$-adenosylmethionine to $S$-adenosylhomocysteine to promote H3K4me3, which is critical to the maintenance of naive pluripotency[64,66,67]. In human ESCs/iPSCs, methionine metabolism replaces threonine metabolism in this function[65]. Which of these two pathways prevails in rabbits iPSCs is unknown.

KEPi cell populations include a subset that expresses the naïve marker CD75 at significantly higher levels than average. These CD75[high] cells lack the typical histone modification of inactive second X chromosome and demonstrate a remarkable capacity for epiblast colonization, resulting in chimeric fetuses, newborns, and viable adults with substantial contributions from iPSCs in every organ analyzed. By compiling data from 10-day-old fetuses, newborns, and young rabbits, we determined a chimeric animal rate of 90% among animals, comparable to rates observed in mice following ESC injection. The degree of chimerism varies widely among organs, ranging from 0.01% to nearly 100%, and also shows significant inter-individual variability within specific organs. This variability likely reflects the stochastic contribution of iPSC-derived progenitor cells to organogenesis rather than a lineage bias of KEPi cells. Importantly, CD75[high] cells exhibit a robust capacity to colonize the germline, as evidenced by the presence of numerous GFP[+] oogonia in the ovaries and successful transmission of the GFP reporter to the next generation. The exceptionally high transmission rate–73% based on proviral DNA integration sites and 100% based on GFP transgenes– suggests near-complete displacement of host germ cells by the injected cells. This observation is surprising, given the relatively low rates of chimerism observed in other organs of female A#6. We hypothesize that the KEP_VALGöX reprogramming protocol primes CD75[high] cells specifically toward the female germline.

These findings establish KEPi_CD75[high] cells as a powerful tool for gene inactivation and the generation of knockout/knock-in rabbits. Further optimization will require developing non-integrative vector systems for KEP expression. A key question is whether transient KEP expression can reprogram iPSCs for germline colonization competence, and if so, whether this competence is stabilized and preserved after transgene extinction.

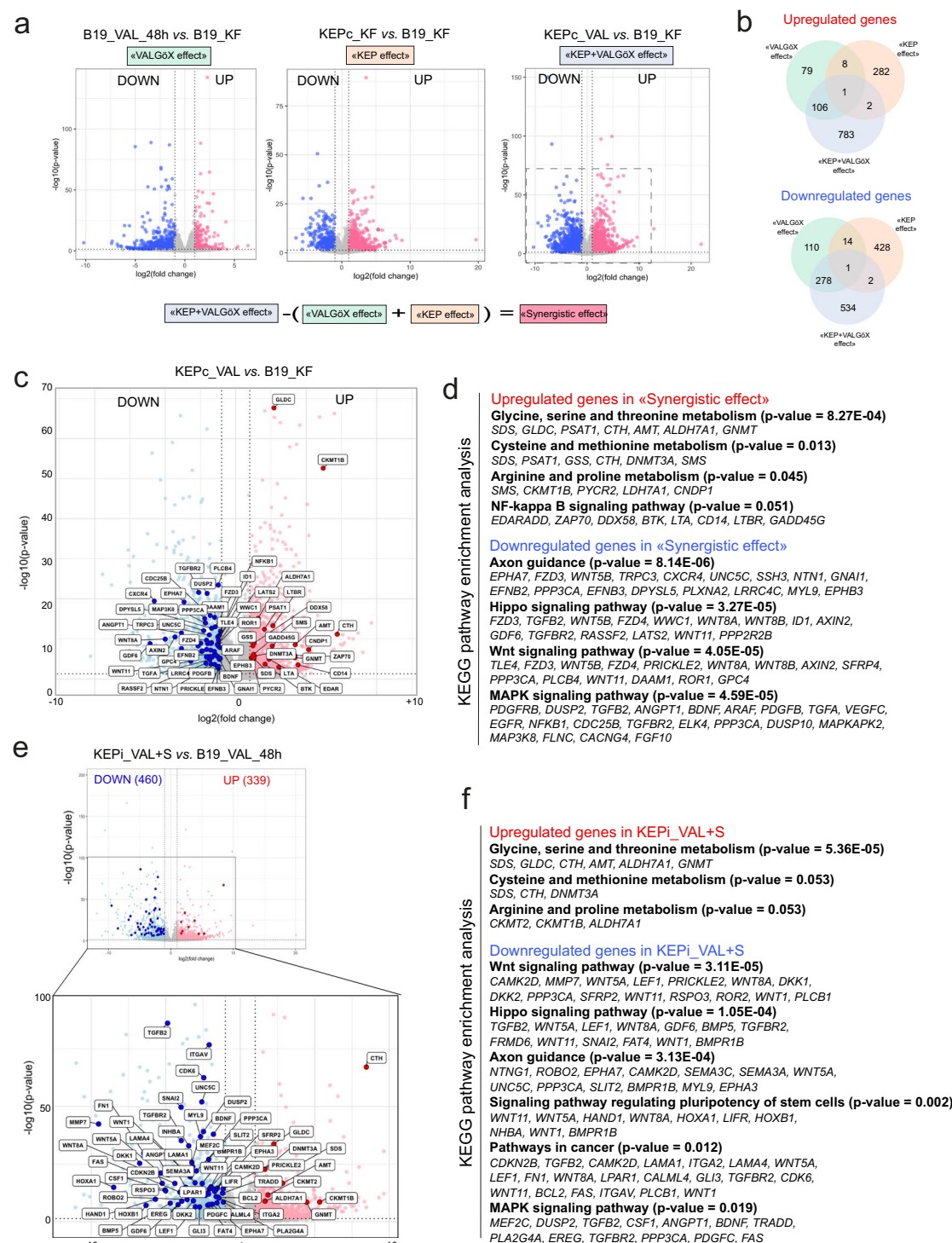

## Methods

### Plasmid constructs

Lentiviral vectors: The simian immunodeficiency virus (SIV)-based vector *pW10-eGFP*[8] was utilized as a backbone vector for generating the cDNA library. All cDNAs (PCR products) were amplified either from plasmids or from reverse-transcribed human PSC RNAs. The origin of cDNAs and the primers for PCR amplification are given in Supplementary Table 3. eGFP was replaced by the cDNAs by subcloning into the *Xho*I restriction site or between the *Afe*I and *Xho*I restriction sites of *pW10-eGFP*.

*DPPA2mKO2:*Two 23-basepair (bp) guide RNA sequences (*5′-CGGCCAGGAGAATGTAAAGATGG-3′*and *5-ACATTATAGCAGCAGCGA-GATGG-3′*) were synthesized(Eurofin). The DNA template for homology-directed repair was constructed in four successive steps.

**Fig. 8 | Transcriptomic signature of iPSCs associated with enhanced embryonic colonization. a** Volcano plot representation of differentially expressed genes (DEGs) in the comparisons between B19_KF and B19_VAL, B19_KF and KEPc_KF, and B19_KF and KEPc_VAL. The dotted box highlights the region magnified in (**c**). **b** Venn diagrams illustrating the overlap of upregulated and downregulated genes among three comparisons: "VALGöX effect", "KEP effect", and the combined "KEP + VALGöX synergistic effect". **c** Enlarged volcano plot from (**a**), showing DEGs between B19_KF and KEPc_VAL. **d** KEGG pathway enrichment analysis of the DEGs associated with the "KEP + VALGöX synergistic effect", highlighting biological pathways enriched among the upregulated and downregulated genes. **e** Volcano plot showing DEGs between B19_VAL and KEPi_VAL + S. **f** KEGG pathway enrichment analysis based on the DEGs identified in (**e**). For all analyses, a Mann–Whitney *U*-test (R base package, version 4.1.2) was used to compare medians between groups. A *p* value <0.05 was considered statistically significant.

First, a 826 bp fragment encompassing the 5′ region of *DPPA2* was amplified from rabbit genomic DNA using primers 5′-*AGGC-CACGTGTCTTGTCCAGAGCTCTCATCTGCTGTATCTGCC*-3′ and 5′-*CATT TTCAGATCGGCCAGGAGAATGTAAAGATGTCAATG*-3′. Second, a 1205 bp fragment containing the coding sequence of the mKO2 fluorescent protein was amplified from pFucci-G₁ Orange (MBL Life Science, AM-V9003M) using primers 5′-*ACATTCTCCTGGCCGATC*-3′ and 5′-*TTAAG GGTTATTGAATATGATCGC*-3′. Third, nested PCR was performed on the 826 bp and 1205 bp fragments using primers 5′-*GTCTTGTCCAG AGCTCTC*-3′ and 5′-*TTAAGGGTTATTGAATATGATCGC*-3′. The resulting 1953 bp fragment was subcloned between the *Sac*I and *Avr*II sites of plasmid *pMA-PGK-neo* (GeneArt LifeTechnologies) to generate *pMA-5′DPPA2-mKO2-PGK-neo-pA*. Finally, the 3′ region of *DPPA2* was amplified from rabbit genomic DNA using primers 5′-*CTTACAGGCG CGCCGAGATGGTAAGTTTTGTTCTAC*-3′ and 5′-*ATCATTGGTACCAGCCT AGGTTAACACTTCC*-3′. The resulting 819 bp fragment was subcloned between AscI and KpnI sites in *pMA-5′DPPA2-mKO2-PGK-neo-pA* to generate *pMA-5′DPPA2-mKO2-PGK-neo-pA-3′DPPA2*.

*EOS^tagBFP*: The double insulator *5′HS4* was amplified from the *pHSC1-HS4*[68] plasmid using primers 5′-*GAATCTGCGGCCTAGAAAGC TTTTTCCCCGTATC*-3′ and 5′-*TAACAGGCCTTAAGCGCTAGCATTGAGC T*-3′ and subcloned into the *Nhe*I site of *PB-loxP-loxP-PGK-Hygro*[69]. Then, a 1593 bp fragment containing the PGK promoter, *neo^r* gene, and polyadenylation signal of the *PGK* gene, was amplified with primers 5′-*A TATATTTTCTTGTTATAGATATCTATCCAACTATGAAACATTA TCATAAG*-3′ and 5′-*CATTATACGAAGTTATGATATCTAATTCTACCGGGTA GGG*-3′ and inserted into the *EcoR*V site of *PB-loxP-loxP-PGK-Hygro* to generate *PB-loxP-loxP-PGK-neo^r*. *TagBFP* was amplified as a 757 bp fragment with primers 5′-*GCGCTTAAGGCCTGTTAACGAATTAGCCATG AGCGAGCTGATTAAG*-3′ and 5′-*GAGGGAGAGGGGCGGAATTCATTACGC CTTAATTAAGCTTG*-3′ and subcloned between the *Hpa*I and *EcoR*I sites. Finally, the *EOS* promoter was amplified with primers 5′-*GCGCTTA AGGCCTGTTAGCCCTCACTCCTTCTCTAGGCGC*-3′and 5′-*CTCATGGCTAA TTCGTTTGTTGCGCCGCCAGCAG*-3′ from plasmid *pL-SIN-EOS-C(3+)-EiP*[26] and inserted into the *Hpa*I site. All construction steps were carried out using the Gibson assembly mix (NEB) and validated by sequencing.

Tagged cDNAs: *HA:ERAS* was amplified from *pW10-ERAS* using primers 5′-*CGTGGAGGAGAACCCCGGCCCCATGACCGAGTACAAGCCCACG-3*′ and 5′-*GATCTAGCTAGCGCCACCATGTACCCATACGATGTTCCAGATTAC GCTATGGAGCTGCCAACAAAGCC*-3′. The resulting 746 bp fragment was subcloned between *Nhe*I and *Hpa*I in *PB-CAG/hygro* to generate *PB-CAG-HA:ERAS-bGHpA-PGKpA-hygro-PGK* (designated as *pCAG-HA:ERAS-hygro*).

*HA:ERAS:DD* was amplified from the *pEX-A128-space-Nhe*I-DD plasmid (Eurofin) using primers 5′-*TTCAATGCTAGCGCCACC*-3′ and 5′-*CCAGGCCCAGGTCGAAGGTGCCAGGCTTTGTTGGCAGCTCCATAGCGTAAT CTGGAACATCGTATGGGTATTCCGGTTTTAGAAGCTCCAC*-3′. The resulting 405 bp fragment was subcloned between *Nhe*I and *Msc*I in *PB-CAG-HA:ERAS-bGHpA-PGKpA-hygro-PGK* to generate *PB-CAG-HA:ERAS:DD-bGHpA-PGKpA-hygro-PGK* (designated as *pCAG-HA:ERAS:DD-hygro*).

*KLF2:V5* was amplified from plasmid *pW10-KLF2* using the primers 5′-*TTCAATGCTAGCGCCACC*-3′ and 5′-*AGCGCTCCTGCAGGCC GCGCTCGCGGCACGGGCTGGCGAAAGTGGAGAAGGACGGCAGGATGG GTTCACTCAGCGCCATTTCCGGTTTTAGAAGCTCCAC*-3′. The resulting 1133 bp fragment was subcloned between *Nhe*I and *Hpa*I in *PB-CAG/neo* to generate *PB-CAG-KLF2:V5-bGHpA-PGKpA-neo-PGK* (designated as *pCAG-KLF2:V5-neo*).

*KLF2:V5:DD* was amplified from *pEX-A128-space-Nhe*I-DD using primers 5′-*TTCAATGCTAGCGCCACC*-3′ and 5′-*AGCGCTCCTGCAGGCC GCGCTCGCGGCACGGGCTGGCGAAAGTGGAGAAGGACGGCAGGATGGG TTCACTCAGCGCCATTTCCGGTTTTAGAAGCTCCAC*-3′. The resulting 406 bp fragment was subcloned between *Nhe*I and *Sbf*I in *PB-CAG-KLF2:V5-bGHpA-PGKpA-neo-PGK* to generate *PB-CAG-KLF2:V5:DD-bGHpA-PGKpA-neo-PGK* (designated as *pCAG-KLF2:V5:DD-neo*).

*Flag:PRMT6* was amplified from plasmid *pW10-PRMT6* using primers 5′- *GATCTAGCTAGCGCCACCATGGATTACAAGGATGACGACGATAA GAGCCCAATGTCGCAGCCCAAGAAAG*-3′ and 5′-*AACTCAGTCCTCCAT GGCAAAG*-3′. The resulting 1175 bp fragment was subcloned between *Nhe*I and *Hpa*I in *PB-CAG/puro* to generate *PB-CAG-Flag:PRMT6-bGHpA-PGKpA-puro-PGK*(designated as *pCAG-Flag:PRMT6-puro*).

*Flag:PRMT6:DD* was amplified by nested PCR from pEX-A128-space-Nhe*I-DD and *pW10-PRMT6* using primers 5′-*TTCAATGCTA GCGCCACC*-3′, 5′-*CTTATCGTCGTCATCCTTGTAATCCATTTCCGGTTTTAG AAGCTCCAC*-3′, 5′-*ATGGATTACAAGGATGACGACG*-3′ and 5′-*TAGTGCT GCTTCACCTGG*-3′. The resulting 1012 bp fragment was subcloned between *Nhe*I and *Oli*I in *PB-CAG-Flag:PRMT6-bGHpA-PGKpA-puro-PGK* to generate *PB-CAG-Flag:PRMT6:DD-bGHpA-PGKpA-puro-PGK* (designated as *pCAG-DD:Flag:PRMT6-puro*).

## Media composition, culture, and electroporation

Mouse embryonic fibroblasts (MEFs) were prepared from 12.5-day-old embryos of OF1 or DR4 mice (Charles River). Mice are housed in groups of five in ventilated cages with bedding, shelter (igloo, tunnel or arch), gnawing material (cotton balls or wooden sticks), and unlimited access to food and drink. The mice are supervised daily by qualified animal keepers. The protocol for MEFs preparation was approved by our research laboratory's animal welfare committee (SBEA). Conventional rabbit iPSCs B19_GFP[8], NaiveRep_KF, and KEP_KF cell lines were routinely cultured on mitomycin C-treated MEFs ($1.6 \times 10^4$ MEFs/cm²) in medium designated as KF comprising Dulbecco's Modified Eagle Medium (DMEM)/F12 supplemented with 1% solution of 10,000 U/mL penicillin, 10,000 U/mL streptomycin, 2 mM L-glutamine, 10 µM β−mercaptoethanol, 20% knockout serum replacement (KOSR), and 10 ng/mL FGF2. The culture medium was replaced every day, and cells were routinely dissociated every 2 or 3 days into single cells by treatment with 0.05% trypsin−EDTA. In NaiveRep_KOSR/LIF cells, FGF2 was replaced by LIF (medium designated as KOSR/LIF). In NaiveRep_FCS/LIF cells, KOSR and FGF2 were replaced by FCS and LIF (medium designated as FCS/LIF). NaiveRep_VAL and KEP_VAL cells were cultured on Matrigel in a medium designated as VALGöX, comprising MEF-conditioned N2B27 basal media supplemented 1% solution of 10,000 U/mL penicillin, 10,000 U/mL streptomycin, 1 mM sodium pyruvate, 10 µM β−mercaptoethanol, 50 µg/mL activin A, 10,000 U/mL leukemia inhibitory factor (LIF), 250 µM Vitamin C, 2.5 µM Gö6983, and 2.5 µM XAV939. The culture medium was replaced every day, and cells were routinely dissociated every 2 or 3 days into single cells by treatment with 0.05% trypsin−EDTA. Shield1 (10 to 1000 nM) was added as indicated.

To establish the NaiveRep cell line, 10⁶ B19_GFP cells were co-transfected using Lipofectamine 2000 (Invitrogen) with 1 µg of *pCAG-Cas9D10A-sgDPPA2(x2)* plasmid and 1 µg of *pMA-5′DPPA2-mKO2-PGK-neo-pA-3′DPPA2* template. G418 (250 µg/mL) was applied for 7 days prior to colony picking and DNA analysis. "On target" integration of the

DNA template was verified by PCR using the following primer pair: *5'-GCAGAGTAAGCCCACTCCAG-3'* and *5'-GACCATCGGCAGGAAAGTTA-3'*. Heterozygote integrations were distinguished from homozygote integrations using the following primer pairs: *5'-GCA-GAGTAAGCCCACTCCAG-3'* and *5'-ACGAGAAAAGCAAGCAGGTC-3'*. In a second step, $10^6$ B19_GFP_DPPA2$^{mKO2}$ cells were co-transfected using Lipofectamine 2000 (Invitrogen) with 1 µg of *pEOS$^{tagBFP}$* plasmid plus 2 µg of the PBase-expressing vector *pCAGPBase*[70]. Puromycin (1 µg/mL) was applied for 7 days to select stable transfectants.

To establish PiggyBac (PB) transgenic lines, $10^6$ cells were co-transfected using Lipofectamine 2000 (Invitrogen) with 1 µg of PB plasmid plus 2 µg of the PBase-expressing vector *pCAGPBase*[70]. Stable transfectants were selected in G418 (250 µg/mL), hygromycin (200 µg/mL), or puromycin (1 µg/mL) for 7 days.

### Virus production, cell infection, and cDNA library screening

About $2 \times 10^6$ 293FT cells (ATCC, ref CRL-3467) were transfected with a DNA mixture containing 4.2 µg of the *pMD2G* plasmid encoding the vesicular stomatitis virus glycoprotein envelope (Addgene, ref #12259), 10 µg of *psPAX2* plasmid encoding the gag, pol, tat, and rev proteins (Addgene, ref #12260), and 12 µg of the *pW10* plasmid carrying the lentiviral genome[71] using the calcium phosphate precipitation technique. The following day, cells were incubated with 5 mL of fresh DMEM and further cultured for 24 h. The supernatant was then collected, filtered (0.8 µm), cleared by centrifugation (3000 rpm, 5 min, 4 °C), and concentrated by ultracentrifugation on 20% sucrose (140,000 × *g*, 2 h, 4 °C).

Prior to infection, $2.4 \times 10^5$ NaiveRep_KF were dissociated with Trypsin−EDTA, and cells were transferred to fresh medium containing viruses at MOI's of 10 to 50, in the presence of 6 µg/mL polybrene. Cells were incubated in suspension for 5 h at 37 °C before being re-plated on fresh feeder cells in KF, FCS/LIF, or KOSR/LIF culture media, or on Matrigel-coated dishes in VALGöX medium, at a density of 100 cells/cm². Colonies showing mKO2 and/or tagBFP fluorescence were picked after 7 to 10 days and expanded.

### Genomic PCR amplification of proviral DNAs

Proviral DNAs were identified by PCR on genomic DNA, either using the Quick-Load Taq2X Master Mix enzyme (NEB, M0271S), or with the Q5 Hot Start High-Fidelity DNA polymerase (NEB, M0493S) and the addition of GC enhancer to aid in the amplification of GC-rich segments. The primer sequences are given in Supplementary Table 4.

### Cell microinjection, embryo culture, and embryo transfer

All procedures in rabbits were approved by the French ethics committee CELYNE (approval numbers APAFIS#6438 and APAFIS#39573). Rabbit embryos were produced by ovarian stimulation. Sexually mature (>6 months old) New Zealand white rabbits were injected with follicle-stimulating hormone and gonadotropin-releasing hormone, followed by artificial insemination with males. A mean number of 6 rabbits was used per experiment, producing ~20 embryos per rabbit. Eight-cell-stage embryos (E1.5) were flushed from explanted oviducts 36−40 h after insemination and cultured in a 1:1:1 mixture of RPMI 1640 medium, DMEM, and Ham's F10 (RDH medium; Thermo Fisher Scientific) at 38 °C in 5% $CO_2$ until cell microinjection. Eight cells were microinjected inside early morula (E2.8) stage rabbit embryos. The embryos were further cultured for 24 h in a 1:1 mixture of cell culture medium and RDH (with or without Shield1). Embryos were then either further cultured in RDH medium (with or without Shield1) after removal of the mucus coat with pronase, or transferred into surrogate mothers. On the day of transfer, eight embryos were transferred to each oviduct of the recipient by laparoscopy. Seven days after transfer, post-implantation embryos (E10.5) were recovered by dissection of the explanted uterine horns. Chimeric females were superovulated using follicle-stimulating hormone and gonadotropin-releasing hormone, followed by artificial insemination with sperm from a wild-type New Zealand white rabbit. Fourteen days later, post-implantation embryos (E14) were retrieved by dissecting the excised uterine horns.

### Immunofluorescence analysis of cells and embryos

Epifluorescent and phase contrast imaging of live cells and chimeric embryos, following injection with GFP$^+$ cells (at the blastocyst stage, 3DIV, or at E10.5 upon collection), was conducted using a conventional fluorescence microscope (TiS; Nikon). This microscope is equipped with DAPI (Ex 377/50, Em 447/60), mKO2 (Ex 546/10, Em 585/40), and GFP (Ex 472/30, Em 520/35) filters. The images were analyzed using NIS-Elements imaging software.

For immunofluorescence analysis, cells and preimplantation embryos were fixed in 4% PFA for 20 min at room temperature. After three washes in phosphate-buffered saline (PBS), they were permeabilized in PBS-0.5% Triton X-100 for 30 min and blocked in 2% BSA for 1 h at room temperature. For 5'methylcytosine immunolabeling, cells were permeabilized in PBS-0.5% Triton for 15 min, washed in PBS for 20 min, then incubated in 2 M HCl for 30 min prior to blocking as above. In all cases, cells and embryos were subsequently incubated with primary antibodies diluted in blocking solution overnight at 4 °C. Primary antibodies include: anti-SOX2 (Bio-Techne, ref AF2018, dilution 1:100), anti-GFP (Invitrogen, ref A10262, dilution 1:200), anti-OCT4 (StemAB, ref 09-0023, dilution 1:200), anti-DPPA5 (R&D systems, ref AF3125, dilution 1:100), anti-OOEP (Abcam ref 185478, dilution 1:100), anti-ubiquityl-Histone H2A Lys119 (Cell Signaling, ref #8240, dilution 1:300), anti-Tri-methyl Histone H3 Lys27 (Cell Signaling, ref #9733, dilution 1:400), anti-Tri-methyl Histone H3 Lys4 (Cell Signaling, ref #9751, dilution 1:400), anti-Histone H3 (acetyl K14) (Abcam, ref ab52946, dilution 1:400), anti-Histone H3 tri methyl K9 (Abcam, ref ab8898, dilution 1:400), anti-Histone H3 (asymmetric di methyl R2) (Abcam, ref ab175007, dilution 1:400), anti-5-methylcytosine (EMD Millipore, ref MABE146, dilution 1:400), and anti-CD75 (Abcam, ref ab77676, dilution 1:1000). After two washes (2 × 15 min) in PBS, they were incubated in secondary antibodies diluted in blocking solution at a dilution of 1:400 for 1 h at room temperature. Secondary antibodies (diluted 1:500) include: donkey anti-goat (Alexa Fluor 555) (Invitrogen ref A21432), donkey anti-chicken (Alexa Fluor 488) (Jackson ImmunoResearch, ref 703-545-155), donkey anti-mouse (Alexa Fluor 647, Invitrogen, ref A21448), donkey anti-rabbit (Alexa Fluor 555, Invitrogen, ref A31572) and donkey anti-rabbit (Alexa Fluor 647; Invitrogen, ref A31573). Finally, they were transferred through several washes of PBS before staining DNA with DAPI (0.5 µg/mL) for 10 min at room temperature. Cells and embryos were analyzed by confocal imaging (DM 6000 CS SP5; Leica). Z-stacks were acquired with a frame size of 1024 × 1024, a pixel depth of 8 bits, and a z-distance of 1 µm between optical sections for cells and 5 µm for embryos. Acquisitions were performed using an oil immersion objective (40x/1.25 0.75, PL APO HCX; Leica) for cells and a water immersion objective (25×/1.25 0.75, PL APO HCX; Leica) for embryos.

### Immunofluorescence analysis of fetuses and tissues

Post-implantation embryos (E10.5) were fixed in 2% paraformaldehyde (PFA) prior to inclusion in OCT and cryo-sectioned. Newborns were dissected and organs were fixed in 2% PFA, embedded in OCT and cryo-sectioned. The chimeric animals were euthanized and perfused via the heart with a 0.9% NaCl solution for 20−30 min to rinse the organs, followed by perfusion with 4% PFA for 45 min. Tissues were then collected and immersed in 4% PFA overnight, after which they were sequentially immersed in 10 and 20% sucrose solutions before being embedded in OCT and cryo-sectioned. Transversal sections, 20-µm thick, were prepared and mounted on Superfrost Plus glass slides (Thermo Scientific) before being stored at −20 °C. For immunostaining, cryosections were air-dried for 30 min and rehydrated in phosphate-buffered saline (PBS) for 30 min. The slides were then

rinsed three times in 0.5% Triton X-100 in PBS and blocked in PBS supplemented with 2% bovine serum albumin (BSA) for 1 h at room temperature. Primary antibodies were incubated overnight at 4 °C in 2% BSA. These included: anti-SOX2 (Bio-Techne, ref AF2018, 1:100), anti-GFP (Invitrogen, ref A10262, dilution 1:200), anti-TUJ1 (Sigma, ref T8660, 1:10000), anti-smooth muscle actin (SMA; Millipore, ref CBL171, 1:500), anti-LAMININ (Sigma, ref I9393, 1:100), anti-PAX3 (DSHB, ref 3929, 1:500), anti-CD31/PECAM-1 (Bio-techne, ref AF806, 1:100), anti-DESMIN (Abcam, ab32362, 1:200), and anti-SOX17 (Bio-Techne, ref AF1924, 1:25). After two washes in PBS, relevant secondary antibodies were incubated in 2% BSA for 1 h at room temperature, at 1:400 dilutions for all: Alexa Fluor 488 goat anti-chicken (Abcam, ab150169), Alexa Fluor 647 donkey anti-mouse IgG (Invitrogen, ref A21448), Alexa Fluor 647 donkey anti-goat IgG (Invitrogen, A21447), and Alexa Fluor 647 donkey anti-rabbit IgG (Invitrogen, ref A31573). Nuclear staining was performed using 4′,6-diamidino-2-phenylindole (DAPI; 0.5 µg/mL in PBS) for 10 min at room temperature. Mounting was done in Vectashield with DAPI (Vectorlabs, H-1200-10). Confocal examination of the fluorescent labeling was conducted on a LEICA DM 6000 CS SP5 equipped with an Argon laser (488 nm), a HeNe laser (633 nm), and a diode at 405 nm. Acquisitions were made using oil immersion objectives (x20 and x40) with the LAS AF software (Leica). Z-stacks were acquired with a frame size of 2048 ×2048 and a z-distance of 1 µm between optical sections.

### Karyotyping of KEP cells
Cells were cultured to ~60% confluence and arrested in metaphase by treating with 0.5 µg/mL KaryoMAX Colcemid (Invitrogen) for 1.5 h. They were then harvested by trypsinization and subjected to hypotonic treatment using a 50:50 mixture of fetal bovine serum (FBS) and 10x KCl solution to induce chromosome swelling. Following two rounds of fixation with cold fixative (3:1 absolute ethanol to acetic acid), the cells are stored at −20 °C for later analysis. For metaphase spreading, 20 µL of the cell suspension was dropped onto Superfrost slides from ~50 cm height and air-dried. Spreads were stained with 1x DAPI for 15 min, mounted using Vectashield®, and examined with a Nikon Eclipse TI epifluorescence microscope at 63× magnification using a DAPI filter (Ex 377/50, Em 447/60). At least 35 metaphases were analyzed for chromosomal abnormalities, with chromosome alignment and karyotype construction performed using Fiji software and the Chromosome J plugin. Chromosome pairs were compared against a reference karyotype to identify structural or numerical anomalies.

### Genomic PCR amplification of transgenes and GFP integration site on rabbit chromosome 1
Blood samples were collected from the ear vessels using heparin vacutainers. Oral cavity specimens were collected by rolling a sterile flocked swab in the mouth, followed by immediate immersion in PBS. E10.5/E14 fetuses and small tissue samples from collected euthanized animals were placed in individual tubes. DNA was extracted using QIAamp DNA mini kit (Qiagen), and concentrations were measured with a Nanodrop 2000. To detect GFP, hKLF2-V5, HA-hERAS, and flag-hPRMT6 and GFP integration site (IS) on chromosome 1, PCR reactions are performed using the Q5® Hot Start High-Fidelity DNA Polymerase kit with specific primers. Each 25 µL reaction contained 500 ng of DNA in an adjusted volume of water, combined with a master mix consisting of 5 µL 5 × Q5 Reaction buffer (with or without 5 µl Q5 GC Enhancer Buffer), 0.5 µL 10 mM dNTPs, 1.25 µL 10 µM forward primer, 1.25 µL 10 µM reverse primer, and 0.25 µL Q5 Hot Start High-Fidelity polymerase. Thermocycling conditions were as follows: initial denaturation at 98 °C for 30 s, followed by 30 cycles of 98 °C, for 10 s, 62 °C for 30 s, and 72 °C for 1 min (for the IS on chromosome 1) or 30 s (for other primer sets), with a final extension at 72 °C for 2 min, and a hold at 4 °C.

PCR products were analyzed by electrophoresis on SYBR safe-stained agarose gels.

For quantitative PCR, samples from euthanized adult chimeras were collected at five different sites within each tissue. DNA was extracted using the QIAamp DNA extraction kit according to the manufacturer's protocol. A standard curve was generated using DNA mixtures of wild-type rabbit embryonic fibroblasts and rabbit B19-GFP. These DNA samples were combined to create a range of GFP percentages (0%, 0.2%, 0.5%, 0.75%, 1%; 5%, 10%, 15%, 20%, 25% 30%, 40%, 60%, and 100%). Real-time PCR was performed using Fast SYBR™ Green Master Mix to detect the presence of the GFP gene and the endogenous TBP gene (used as a normalizing control). GFP percentages were calculated by normalizing to the TBP gene and using the standard curve.

### Determination of genomic integration sites by Ligation-mediated PCR (LM-PCR)
LM-PCR was performed following the protocol of Ciuffi et al.[72] with modifications. Genomic DNA was extracted from KEPi cells using the Qiamp DNA extraction kit (Qiagen, #51304) and digested overnight with MseI (NEB, #R0525S). The resulting DNA fragments were ligated overnight using T4 DNA ligase and 20 µM linkers [MseI linker (+): 5′-GTAATACGACTCACTATAGGGCTCCGCTTAAGGGAC-3′; MseI linker (-): (Phosp)TA GTCCCTTAAGCGGAG(Amino Modifier C6)]. To remove internal and 5′ fragments, linker-ligated DNA was further digested with either SacI (NEB, #R3156S) or KpnI (Thermo Scientific, # FD0524). The first round of PCR was carried out using the MseI primer (5′-GTAATACGACTCACTATAGGGC-3′) and GAE primer-1 (5′-AGTAAGCCAGTGTGTGT TCC-3′), with Accuprime™ GC-rich DNA Polymerase (Thermo Fischer, #12337024) under the following conditions: initial denaturation at 95 °C for 3 min, followed by 30 cycles of denaturation (95 °C, 30 sec), annealing (60 °C, 1 min), and extension (72 °C, 1 min), with a final extension at 72 °C for 10 min. The primary PCR products were then digested with HincII (NEB, #R0103S) and amplified by nested PCR using the MseI nested primer (5′-AGGGCTCCGCTTAAGGGAC-3′), GAE primer-2 (5′CGGTAATAAGAAGACCCTGGTC-3′), and GAE primer-3 (5′-AGGACCCTTTCTGCTTTGAG-3′). Nested PCR products were ligated into pJET1.2 plasmid (Thermo Fischer, #K1232), transformed into 10-beta Competent E. coli (NEB, #C3019), and colonies were screened by PCR with pJET1.2-for (5′-CGACTCACTATAGGGAG-3′) and pJet1.2-rev (5′-ATCGATTTTCCATGGCAG-3′) primers and the Quick-Load® Taq (NEB) according to manufacturer instructions. Plasmid DNA was extracted from colonies and integration sites were directly sequenced (Microsynth) using the pJET1.2 reverse primer. Successfully cloned integration sites are mapped to the rabbit genome using BLAT (UCSC GenomeBrowser, oryCun2, April 2009) and BLAST (NCBI, mOryCun 1.1).

### Flow cytometry analysis and sorting
Cells were dissociated to a single cell suspension with 0.1% trypsin, and incubated in 10% fetal bovine serum for 15 min at room temperature for epitope blocking. Cells were subsequently incubated with CD75 primary antibody (Abcam, ref ab77676; dilution 1:50) for 30 min at room temperature, followed by goat anti-mouse IgG secondary antibody coupled to Alexa Fluor™ Plus 647 (Invitrogen, ref A32728; dilution 1:200) for 30 min. After several washes in PBS, cells were analyzed using LSRFortessa™ X-20 Cell Analyzer (Beckton Dickinson), and the data were analyzed using BD FACSDiva™ software. The CD75$^{high}$ subpopulation was sorted using FACSAria™ III Sorter (Beckton Dickinson). Total cells were gated based on FSC-A and SSC-A. Single cells were selected using FSC-H/FSC-A gating. "No antibody" and "no primary antibody" controls were employed to identify any unspecific labeling.

## Western blotting

Cells were lysed in cold RIPA lysis buffer (0.5% NP-40, 1% Triton X, 10% glycerol, 20 mM Hepes pH 7.4, 100 mM NaCl, 1 mM sodium orthovanadate, 0.1% DTT, and protease inhibitors (Roche, ref #05 892 970 001) for 4 h at 4 °C. Lysates were cleared by centrifugation for 15 min and stored at −80 °C. Protein concentrations were measured using the Bradford assay. For SDS-PAGE electrophoresis, 30 μg of total proteins were loaded onto each well of Mini-PROTEAN TGX Stain-Free Precast Gels (10%, Biorad, ref # 4568031), and migrated for 45 min at 120 Volts. Precision Plus Protein Dual Color Standards (Biorad, ref # 1610374) were used as a protein ladder. After electroporation, proteins were transferred onto membranes using Trans-Blot® Turbo™ RTA Midi 0.2 μm Nitrocellulose Transfer Kit (Biorad, ref #1704271). The membranes were subsequently blocked in TBST solution (200 mM Tris-HCl, 1.5 M NaCl, 0.1% Tween-20, 5% milk) for 1 h at RT, prior to incubation with primary antibodies diluted in TBST at 4 °C for 12 h. Primary antibodies include: anti-V5 Tag (Invitrogen, ref R96025, 1:500, anti-HA (Sigma-Aldrich, ref H6908, 1:500), anti-Akt (pan) (Cell Signaling, ref #4691, 1:1000), anti-Phospho-Akt (Ser473) (Cell Signaling, ref #4060, 1:1000), and anti-beta-actin (Sigma-Aldrich, ref A3854, 1:10000). Membranes were incubated with HRP-conjugated secondary antibody (Jackon ImmunoResearch anti-rabbit ref 211-032-171 and anti-mouse ref 115-035-146, dilution 1:5000) for 1 h at RT. After serial washing in TBST, HRP activity was revealed using Clarity™ Western ECL substrate (Biorad, ref #170-5060) and ChemiDoc™ MP imaging system (Biorad).

## RNA extraction, RT-qPCR, and RNA sequencing

For RT-qPCR, total RNA was extracted using the RNeasy mini kit (Qiagen ref #74106). cDNA was synthesized from 500 ng total RNA using the High-Capacity RNA-to-cDNA Kit (Invitrogen, 4387406) according to the manufacturer's protocol. Quantitative PCRs were performed using Fast SYBR™ Green Master mix (Applied Biosystems™, 4385612) and the Step One plus Real-time PCR system. TBP gene were used as reference. Data were analyzed using comparative CT (ΔΔCT) - quantitative method.

For RNA sequencing, total RNA was isolated using the trizol/chloroform protocol followed by RNeasy mini kit (Qiagen ref #74106) with a DNase I (Qiagen, ref #79254) treatment. Three nanograms of total RNA were used for amplification using the SMART-Seq V4 Ultra Low Input RNA kit (Clontech) according to the manufacturer's recommendations (ten PCR cycles were performed). cDNA quality was assessed on an Agilent Bioanalyzer 2100, using an Agilent High Sensitivity DNA Kit. Libraries were prepared from 0.15 ng cDNA using the Nextera XT Illumina library preparation kit. Libraries were pooled in equimolar proportions and sequenced (Paired-end 50–34 bp) on an Illumina NextSeq500 instrument, using a NextSeq500 High Output 75 cycles kit. Demultiplexing was performed using bcl2fastq2 (version 2.18.12) and adapters were trimmed with Cutadapt (version 1.15 and 3.2) so that only reads longer than 10 bp were kept. Number of reads ranged from 10 to 200 million post adapter trimming. Reads were mapped to the rabbit genome (*OryCun2_ensembl92*) using tophat2. 49.8 to 58.4% (depending on samples) of the pair fragments could be uniquely mapped to the gene reference.

## Image analysis

Fluorescent profiles measurements and quantitative image analysis were conducted using Fiji. Briefly, after z-projection of the stack (sum slices function), nuclei were first segmented using an automatically set threshold and watershed separation. Signals to be quantified were then segmented with the masks defined by the nuclei. For segmenting the spot corresponding to the X chromosome, the size and the circularity of the objects to be detected within each nucleus were determined with the control group. The same parameters were then used on the other groups to determine how many spots could be detected per cell. For all detected objects, the area and mean intensity were calculated and exported for statistical analysis.

## Bioinformatics analysis

Each sample was analyzed in three biological replicates, produced at least 1 week apart. For bulk RNA-seq, count tables were generated using FeatureCounts (version 1.5.0-p2). All analyses were executed using R software (version 4.1.2). Data normalization, gene expression levels, and PCA were performed using the DESeq2 R package (version 1.34.0)[73]. Mean expression levels and standard deviations were calculated using the R base package (version 4.1.2) for each sample. After log2 transformation, normalized counts were used to produce heatmaps with the pheatmap R package (version 1.9.12). DESeq2 computed log2 transformed fold-change (FC) and *p* values for each feature. Differentially expressed genes (DEGs) were determined by Over-Representation Analysis (ORA) using DESeq2 results and the following thresholds: FC >2 ∪ FC <(−2), *p* value < 0.01 or <0.05 as indicated. Venn diagrams were produced using the VennDiagram package (version 1.7.3) and volcano plots were designed using the ggplot2 package (version 3.4.1). Finally, KEGG (Kyoto Encyclopedia of Genes and Genomes) pathway enrichment analysis was performed using the EnrichR[74] web tool and the human KEGG pathways database (release 2021). Gene expression analysis at the single-cell level in rabbit embryos was carried out using data from ref. 24 (GSE180048). The dataset was processed using the Seurat R package (version 4.3.0). Bulk RNA-seq data from rabbit embryos were retrieved from ref. 24 (PRJNA743177) and were analyzed using the DESeq2 R package.

## Quantification and statistical analysis

The Mann–Whitney *U*-test, suitable for non-normal distribution was conducted using the R base package (version 4.1.2) to compare medians between samples. A difference was considered significant if the *p* value associated with the difference between medians was less than 0.05. For immunofluorescence analysis, either conventional *T*-test or Welch's unequal variances *t*-test (two-sided) was employed. A Shapiro–Wilk test was performed to evaluate normality (high *p* values). The SuperPlotsOfData tool[75] was utilized to account for all measurements and biological replicates in these comparisons.

## Reporting summary

Further information on research design is available in the Nature Portfolio Reporting Summary linked to this article.

# Data availability

RNA-seq data have been deposited at GEO with accession number GSE250288. The confocal image data are available under restricted access due to technical limitations related to data volume and storage infrastructure. Access can be obtained by submitting a request to the corresponding authors. Upon approval, the data will be transferred using a secure file-sharing service within 1 week of the request. The data will remain accessible for the duration of the research project or up to one year after access is granted. Source data are provided with this paper.

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

## Acknowledgements

We would like to thank Guillaume Marcy for his invaluable help with the bioinformatic analysis; Pascale Giroud and Dorothé Pattie for their help with animal euthanasia, perfusion, and histological analysis; Manon Dirheimer, Angèle Bellemin-Ménard, Flavia Couto, Julie Lachaud, Lucie de Labuxière, and Jérémy Lavignas for expert animal care and surgery. This work has benefited from the facilities and expertise of the high-throughput sequencing core facility of I2BC (Yan Jaszczyszyn, Magali Perrois, Centre de Recherche de Gif, http://www.i2bc.paris-saclay.fr/), the cytometry core facility of CRCL (Priscillia Battiston-Montagne, Thibault Andrieux, Centre de Recherche en Cancérologie de Lyon, https://www.crcl.fr/en/platforms/flow-cytometry-core-facility/), and the staff members of the animal facility of SBRI (Stem Cell and Brain Research Institute, https://sbri.fr/). This work was supported by the Agence Nationale pour la Recherche (contracts ANR-18-CE13-023, ORYCTO-CELL; ANR-21-CE20-0018-01, CHROMNESS), the Fondation pour la Recherche Médicale (DEQ20170336757 and EQU202303016295 to P.S.), the Infrastructures Nationales en Biologie et Santé (ANR-11-INBS-0009, INGESTEM; CRB-Anim, ANR-11-INBS-0003), the IHU-B CESAME (ANR-10-IBHU-003), the LabExs (ANR-10-LABX-73, REVIVE; ANR-10-LABX-0061, DEVweCAN; ANR-11-LABX-0042, CORTEX), and the University of Lyon within the program "Investissements d'Avenir" (ANR-11-IDEX-0007).

## Author contributions

Investigation: H-T.P., F.P., Y.P., N.D., S.R-G., M.G., A.M., M.R., E.D.S.F., V.B., F.W., B.P., I.P., M.A. and N.B.; Bioinformatic analysis: Y.P., L.J. and V.D.; Writing original draft and funding acquisition: P.S. and N.B.; Conceptualization, supervision, validation, visualization, project administration, manuscript review and editing: P.S., M.A., N.B.; Resources: T.J.

## Competing interests

The authors declare no competing interests.
