## [Transparent Peer Review file · Nature Communications]

Efficient Generation of Germline Chimeras in a Non-Rodent Species Using Rabbit Induced Pluripotent Stem Cells

Corresponding Author: Dr Pierre Savatier

Version 0:

Reviewer comments:

Reviewer #1

(Remarks to the Author)

In this study Perold and colleagues report the generation and propagation of rabbit induced pluripotent stem cells that are capable of substantial and widespread contributions to liveborn rabbits. The results are clear and largely sufficient. Although germ line colonisation has not been demonstrated, high levels of fetal chimerism and liveborn chimeric animals is a major advance which has previously only been proven for mice and rats. I have two suggestion for additional results that would improve the study. The second of these may be considered optional.

1. Viable liveborn chimeras is the standout result from this study. However, the evidence of chimerism should additionally include quantitative analysis of blood samples for GFP by flow cytometry, and ideally also analysis of sections of a tissue biopsy e.g. an ear punch.

2. The authors could clarify the signaling milieu required to maintain the rabbit iPSCs including whether they can withstand inhibition of the FGF/ERK pathway, which is considered a defining characteristic of naïve pluripotency in rodents and human. The manuscript begins by reporting a medium VALGöX established during trials for conversion of primed-type iPSCs. The authors then use this medium in a reprogramming setting to derive expandable chimaera-competent iPSCs. As far as I can follow from the manuscript, the authors do not test whether all components of VALGöX are necessary to maintain the reprogrammed cells. Notably, LGöX are components of the media used for human naïve PSCs, with the addition of a MEK/ERK inhibitor. It should be straightforward and fast for the authors to determine whether their cells are dependent on activin and FGF signaling, which would bracket them with formative/primed pluripotent stem cells, or do not require FGFR or MEK/ERK signaling, a unique feature of rodent and human naïve PSCs. In addition, VALGöX appears to include MEF conditioned medium – is this necessary? It is also not clear whether the reprogrammed cells are maintained on feeders or on geltrex. This information would be of great interest and value for the field.

Other comments

Introduction

- Systemic chimerism and germline chimerism are not the same, even in mice.
- Overall I find the Introduction dwells too much on questionable claims of totipotent cells and chimeric macaques. It is also not appropriate to refer to the recent report of chimeric macaques as “groundbreaking” until those results are independently verified. More discussion of what has previously been found in rabbit and livestock mammals would make a more balanced introduction.
- The Intro would also benefit from a brief discussion of the reporter and transgene approaches previously used to convert mouse and human primed PSCs to naïve status, because these are direct precedents for the approach used here.

Results

- Overall, pages 4-10 of Results and associated figures could be streamlined to make the paper and key findings more accessible.
- Please begin by summarizing how B19 cells were made and whether they are free of reprogramming transgenes.
- What is OOEP and why is it a naïve marker (I am not aware of it in mouse or human).
- A decrease in H3K27me3 is not evident in Fig 1B, and moreover is not an expectation of naïve pluripotency from mouse or human both of show overall increased H3K27me3 in naïve PSCs (van Mierlo, 2019; Zijlman, 2022).
- The authors likely have examined fetal gonads for evidence of colonization. It would be instructive to hear the findings, though I do not consider that germline contribution should be a criterion for publication.
- Lack of H3K27me3 and H2AUb foci suggests but does not prove XaXa status. That requires demonstration of biallelic expression.
- DNMT3B is upregulated in primed compared with naïve PSCs in both mouse and human, so why is it included in the list of

naïve markers. DNMT3L is up-regulated in naïve PSCs.

- Figure 4B would be more accessible split into two separate plots for Nos of chimaeric embryos and Nos of GFP cells.
- Shield acts post-transcriptionally therefore the effect of Shield withdrawal should not be described as silencing.
- Not all of the blastocyst stage chimaera figures are required and overall they detract from the more important and compelling fetal chimaera results.
- Apparent location of GFP labelled cells in the outer layer of the blastocyst is not sufficient to assert trophoblast colonization without demonstrating co-expression of trophoblast determinants such as CDX2 and GATA3.
- The text and/or figure legends should state explicitly that images in Fig 6G and 7D show GFP antibody staining, ruling out autofluorescence artefacts that have beset other studies.
- Please indicate the expected recovery rate of embryos transferred without manipulation.

Discussion

- Genomic PCR on the placenta is not sufficient to assert trophoblast contribution because the signal could be derived from mesoderm derivatives. The authors should be cautious on the issue of expanded potency (which is not defined!)
- The discussion would benefit from describing expression of ERAS, PRMT6 and KLF2 in the early rabbit embryo.
- It is somewhat misleading to compare the observed chimerism with that for mouse ES cells. The frequency of fetuses and term animals with detectable contributions may be similar, but the extent of colonization is clearly less than can be routinely obtained in mouse or rat.
- Gene invalidation is not correct terminology

Reviewer #2

(Remarks to the Author)

In the article "Viable chimeric rabbits with a high contribution of induced pluripotent stem cells", the authors have pioneered a novel method for generating chimeric rabbits by refining the culture medium and overexpressing specific genes. The study's findings are substantial in their potential to advance rabbit use in laboratory research. Although germline chimeric rabbits were not generated, the investigation is compelling nonetheless. However, prior to considering publication, I would like to see the following revisions:

1. As described in the manuscript, rabbit embryonic stem cells (ESCs) cultured in KF medium exhibit a primed state of pluripotency. However, this finding is intriguingly at odds with the observation that NaiveRep_KF cells express the naive-specific reporter EOS-TagBFP. To reconcile these seemingly contradictory results, the authors should provide further explanation or discussion in the text (Fig. 1g).
2. For the four rounds of screening, it is notable that Screen 1 differs from the other three screenings (Screens 2a, 2b, and 3). Specifically, the cDNA library used in Screen 1 contains only 25 genes, whereas Screens 2a, 2b, and 3 utilize a library with 36 genes. Additionally, the multiplicity of infection (MOI) for Screen 1 is 50, whereas it is 15 for the other screens. These discrepancies warrant further explanation or discussion in the text to clarify their impact on the results.
3. The statements "This indicates that transgene silencing shifted KEPI_VAL cells away from naïve-like pluripotency" and "Nevertheless, the Shield1-deprived cells continued to self-renew without any apparent differentiation" appear to be contradictory. The first statement suggests that transgene silencing leads to a loss of naive-like pluripotency, while the second statement implies that the cells remained in a state of self-renewal without differentiating. This possible inconsistency should be addressed or clarified in the text.
4. The manuscript reports on the presence of CD75^{high} cells in KEPI#13_VAL+S and KEPI#28_VAL+S. Specifically, what is the percentage of CD75^{high} cells in these populations? Furthermore, are the percentages of CD75^{high} cells different between Condition I (24-hour culture) and Condition II (5-day culture)?
5. In Figure 7a, the electrophoretic image appears to show two distinct lines corresponding to muscles. Are these two lines representative of different kind of muscles?
6. Can the GFP gene be detected in germline-related cells such as Sertoli cells, sperm, oocytes, or cumulus cells?
7. For a stably established cell line, karyotype consistency is crucial. It is essential to demonstrate that the cells cultured in VAL with overexpressed KEP exhibit a stable chromosomal count and structure.
8. It would be highly valuable to derive naive rabbit ESCs without genetic modification. Is it feasible to achieve this goal? A more in-depth discussion on the possibility and potential approaches for accomplishing this feat would be beneficial.

Reviewer #3

(Remarks to the Author)

These authors successfully generated viable chimeric rabbits using pluripotent stem cells. It is a great work and experiments were carefully performed. Importantly, they identified three genes—KLF2, PRMT6, and ERAS—that enable rabbit iPSCs to acquire molecular and functional features of the naïve state including the capacity to efficiently colonize preimplantation embryos. This work should be congratulated but the paper can be improved by providing the discussion on the possible germline transmission and potential application of this rabbit model for the research in biomedical field in the discussion section.

Version 1:

Reviewer comments:

Reviewer #1

(Remarks to the Author)

In their revised manuscript Perold and colleagues provide constructive responses with additional data to address the key issues raised by reviewers. Indeed, they go beyond what was strictly required to show evidence of germline transmission from rabbit iPSCs. This is a potentially a landmark result that merits high profile publication, but must be demonstrated beyond doubt. I therefore hope the authors can address a major concern with the data presented.

Figure 7E shows the results of genomic PCR assay for GFP in foetuses from litters of the two female chimaeras. Every foetus (27/27) shows a strong signal for the GFP transgene. This is a quite extraordinary result that requires further explanation in two respects. Firstly, this result would mean that the iPSCs have completely displaced host germ cells. What proportion of oogonial cells in the ovaries of these two animals were GFP+ve (the text says numerous, not all; images are provided but no quantitation)? Can the authors explain how could germline chimaerism reach 100% in animals that show very low contributions in most tissues (Rabbit 6 is <0.5% in all tissues except lung). Secondly, even if all the oocytes are iPSC derived, they would not all be expected to transmit the GFP transgene due to segregation during meiosis. We are not told how many copies of the transgene are in the B19-GFP cells, but there would have to be several independent integrations to explain this result. Furthermore, in such a case the copy numbers would vary between the offspring, which is not evident from the PCR gel. Please clarify and provide evidence to justify. Moreover, epifluorescence images are not definitive without showing non-transgenic foetuses in the same image and/or corroboration by GFP antibody staining as elsewhere in the paper. In addition, information should be provided whether all foetuses have normal anatomy and developmental staging. Finally, there is a discrepancy between the number of germline foetuses declared in the text, 25, and the 27 samples shown in Fig 7E.

Other points:

The revised Introduction is much improved. The statement on lines 69-71 that "In mice, this has been achieved by reprogramming EpiSCs through the overexpression of specific transcription factors, including Klf4, Nr5a1, and Nr5a2, in culture conditions with LIF, MEK inhibitor PD0325901, and GSK3 inhibitor CHIR99021, known as 2iLIF" is somewhat misleading, however. Germline transmission was first shown for mouse ES cells by Bradley et al in 1984 and 2iLIF culture was developed for naive ES cells without use of transgenes. I suggest those points should be made before EpiSC reprogramming.

The Results text should state when first describing VALGoX medium that this is in MEF-conditioned medium (which they later show is required).

In Table 2 it is reported that chimaera A2 has 100% iPSC contribution by quantitative gPCR analysis, but the image of skin from that animal in Fig 7C shows a large number of GFP-ve cells. This difference should be discussed.

Line 524: "These CD75high cells have reactivated the second X chromosome" is over-interpreting the histone modification staining as raised in the previous review.

In Table 1 there is some mislabelling in the Sex and Name column

Reviewer #2

(Remarks to the Author)

The manuscript entitled "Efficient Generation of Germline Chimeras in a Non-Rodent Species Using Rabbit Induced Pluripotent Stem Cells" has addressed all the concerns I raised in the previous review. I believe the revised version is sufficiently polished, and recommend it for publication.

Reviewer #3

(Remarks to the Author)

The paper has been successfully revised.

Version 2:

Reviewer comments:

Reviewer #1

(Remarks to the Author)

The authors have addressed one of the critiques adequately but the response to the second is insufficient and seems unnecessarily rushed.

Germline colonisation: They have quantified the high frequency of GFP+ oogonia and provided a speculative but plausible explanation for the near complete (ca 90%) colonisation of the female germline. I agree that providing a mechanistic analysis is beyond the scope of this paper.

Transgene transmission: They provide a new gPCR analysis using reduced template DNA that shows different intensities of GFP band. However, there is no loading control provided for this analysis, nor any copy number titration. More fundamentally, the number of integrations in the B19 line is assumed to be multiple, but this is not evidenced. Integration site sequence analysis should be performed to quantify the number of integrations and to demonstrate inheritance of different subsets. I recognise that I did not specify this assay in my review, but it is a reasonable expectation. There should not be a technical problem in performing integration analyses on the DNA samples the authors have in hand. The observation of

100% transgene transmission is so unusual and the proof of germline competence for a non-rodent embryonic stem cell so significant for the wider field, that there should be no room left for doubt.

Version 3:

Reviewer comments:

Reviewer #1

(Remarks to the Author)

The authors have provided the data necessary to confirm beyond reasonable doubt germline transmission of the GFP transgene. I commend the study.

Reviewer #1 (Remarks to the Author):

In this study Perold and colleagues report the generation and propagation of rabbit induced pluripotent stem cells that are capable of substantial and widespread contributions to liveborn rabbits. The results are clear and largely sufficient. Although germ line colonisation has not been demonstrated, high levels of fetal chimerism and liveborn chimeric animals is a major advance which has previously only been proven for mice and rats. I have two suggestions for additional results that would improve the study. The second of these may be considered optional.

1. Viable liveborn chimeras is the standout result from this study. However, the evidence of chimerism should additionally include quantitative analysis of blood samples for GFP by flow cytometry, and ideally also analysis of sections of a tissue biopsy e.g. an ear punch.

Response: In the revised manuscript, we have addressed the reviewer's request providing a substantial amount of new data. Specifically, we sacrificed the chimeric founders to examine chimerism rates across various organs including the gonads using both quantitative genomic PCR and tissue section analysis. Additionally, we analyzed the progeny of the three female founders, observing germline transmission of the iPSC genome in two of them. These new results are reported in Table 2, detailed on page 13-14 (lines 387-407), and illustrated in Figs. 7 and S12 of the revised manuscript.

2. The authors could clarify the signaling milieu required to maintain the rabbit iPSCs including whether they can withstand inhibition of the FGF/ERK pathway, which is considered a defining characteristic of naïve pluripotency in rodents and human. The manuscript begins by reporting a medium VALGÖX established during trials for conversion of primed-type iPSCs. The authors then use this medium in a reprogramming setting to derive expandable chimaera-competent iPSCs. As far as I can follow from the manuscript, the authors do not test whether all components of VALGÖX are necessary to maintain the reprogrammed cells. Notably, LGÖX are components of the media used for human naïve PSCs, with the addition of a MEK/ERK inhibitor. It should be straightforward and fast for the authors to determine whether their cells are dependent on activin and FGF signaling, which would bracket them with formative/primed pluripotent stem cells, or do not require FGFR or MEK/ERK signaling, a unique feature of rodent and human naïve PSCs. In addition, VALGÖX appears to include MEF conditioned medium – is this necessary? It is also not clear whether the reprogrammed cells are maintained on feeders or on geltrex. This information would be of great interest and value for the field.

Response: In the revised manuscript, we have included new data addressing the reviewer's concerns. Specifically, we demonstrate that the withdrawal of each component of the VALGÖX medium from KEP cell cultures leads to differentiation within 21 days. These findings are detailed on page 9 (lines 245-250) and illustrated in Fig. S4. Additionally, the reprogrammed cells are maintained on feeders as specified in the Result section (page 4, line 91) and Materials & Methods section (page 21, lines 628-630).

Other comments

Introduction

- Systemic chimerism and germline chimerism are not the same, even in mice.

Response: In the revised manuscript, we have removed references to “systemic chimerism” for clarity.

- Overall, I find the Introduction dwells too much on questionable claims of totipotent cells and chimeric macaques. It is also not appropriate to refer to the recent report of chimeric macaques as “groundbreaking” until those results are independently verified. More discussion of what has previously been found in rabbit and livestock mammals would make a more balanced introduction. The Intro would also benefit from a brief discussion of the reporter and transgene approaches previously used to convert mouse and human primed PSCs to naïve status, because these are direct precedents for the approach used here.

Response: In the revised manuscript, we have rewritten the Introduction to align with the reviewer's suggestions. It now includes a more balanced discussion of existing studies in rabbits and livestock, as well as an overview of previous reporter and transgene approaches for converting mouse and human primed PSCs to naïve status see pages 3 & 4.

Results

- Overall, pages 4-10 of Results and associated figures could be streamlined to make the paper and key findings more accessible.

Response: In the revised manuscript, we have streamlined the Results section on pages 4-10, as suggested by the reviewer, to improve clarity and accessibility.

- Please begin by summarizing how B19 cells were made and whether they are free of reprogramming transgenes.

Response: In the revised manuscript, we have updated the beginning of the Results section (page 4, lines 87-89) to include a description of the female B19 iPSC line. The B19 cells were generated using retroviral vectors encoding human OCT4, SOX2, KLF4, and c-MYC. Following reprogramming, all four transgenes were fully silenced, as reported in Osteil et al. (Osteil et al. 2013).

Osteil P, Taponnier Y, Markossian S, Godet M, Schmaltz-Panneau B, Jouneau L, Cabau C, Joly T, Blachere T, Gocza E et al. 2013. Induced pluripotent stem cells derived from rabbits exhibit some characteristics of naive pluripotency. Biol Open 2: 613-628.

- What is OOEP and why is it a naïve marker (I am not aware of it in mouse or human).

Response: OOEP is structurally related to DPPA5. We identified it as a naïve marker in our previous study on rabbit embryos (Bouchereau et al. 2022).

Bouchereau W, Jouneau L, Archilla C, Aksoy I, Moulin A, Daniel N, Peynot N, Calderari S, Joly T, Godet M et al. 2022. Major transcriptomic, epigenetic and metabolic changes underlie the pluripotency continuum in rabbit preimplantation embryos. Development 149: dev200538.

- A decrease in H3k27me3 is not evident in Fig 1B, and moreover is not an expectation of naïve pluripotency from mouse or human both of show overall increased H3K27me3 in naïve PSCs (van Mierlo, 2019; Zijlman, 2022).

Response: We agree with the reviewer's observation. In the revised manuscript, we have removed the H3K27me3 immunostaining from panel 1b to address this concern.

- The authors likely have examined fetal gonads for evidence of colonization. It would be instructive to hear the findings, though I do not consider that germline contribution should be a criterion for publication.

Response: We examined the gonads of our three adult female chimeras and observed strong colonization within the ovarian follicles of two of them. Additionally, we analyzed the progeny of these three female founders and confirmed germline transmission of the iPSC genome for two of them (the third gave no offspring). Notably, the rate of transmission was 100%, suggesting that the germline of these two chimeras is entirely derived from the iPSCs. These new findings are detailed on page 13-14 (lines 387-407) and illustrated in Figs. 7 and S12 of the revised manuscript.

- Lack of H3K27me3 and H2AUb foci suggests but does not prove XaXa status. That requires demonstration of biallelic expression.

Response: We acknowledge this oversight. In the revised manuscript, we corrected the error on page 9 (line 242) to clarify that the absence of these foci only suggests, rather than proves, an XaXa status.

- DNMT3B is upregulated in primed compared with naïve PSCs in both mouse and human, so why is it included in the list of naïve markers. DNMT3L is up-regulated in naïve PSCs.

Response: We recognize the error and have corrected it in the revised manuscript. *DNMT3B* was mistakenly listed as a naïve marker on page 9 (line 258), and this has been revised.

- Figure 4B would be more accessible split into two separate plots for Nos of chimaeric embryos and Nos of GFP cells.

Response: In the revised manuscript, we have split Figure 4b into two separate histograms: Figure 4b now shows the number of chimeric embryos, and Figure 4c displays the number of GFP-positive cells.

- Shield acts post-transcriptionally therefore the effect of Shield withdrawal should not be described as silencing.

Response: We have corrected this terminology in the revised manuscript. The term "silencing" has been replaced to reflect the post-transcriptional effect of Shield withdrawal, as indicated on pages 11 (lines 311, 312, and 323) and 15 (line 435) of the revised manuscript.

- Not all of the blastocyst stage chimaera figures are required and overall they detract from the more important and compelling fetal chimaera results.

Response: We agree with the reviewer's suggestion. In the revised manuscript, portions of Figure 4a, 5e, and 6c have been moved to supplementary Figures S5, S7, and S9, respectively.

- Apparent location of GFP labelled cells in the outer layer of the blastocyst is not sufficient to assert trophoblast colonization without demonstrating co-expression of trophoblast determinants such as CDX2 and GATA3.

Response: We acknowledge this limitation. We did not observe GFP+/GATA3+ cells after double immunostaining of chimeric embryos. As a result, we have removed the term "trophoblast colonization" from the revised manuscript.

- The text and/or figure legends should state explicitly that images in Fig 6G and 7D show GFP antibody staining, ruling out autofluorescence artefacts that have beset other studies.

Response: Thank you for pointing out this issue. We have updated the figure legends for Fig. 6g (S9b and S9c in the revised ms) and 7d (S11c in the revised ms) to explicitly state that these images show GFP antibody staining. Additionally, GFP is indicated on the images to avoid confusion.

- Please indicate the expected recovery rate of embryos transferred without manipulation.

Response: The recovery rate of chimeric fetuses transferred (17%) is similar to that observed in two studies with unmanipulated embryos [27% in Besenfelder et al. (1993); 9.3% in Besenfelder et al. (1998)]. This information is now included on page 13 (lines 364-365) of the revised manuscript.

Besenfelder, U. & Brem, G. Laparoscopic embryo transfer in rabbits. J Reprod Fertil 99, 53-56 (1993). <https://doi.org/10.1530/jrf.0.0990053>

Besenfelder, U., Strouhal, C. & Brem, G. A method for endoscopic embryo collection and transfer in the rabbit. Zentralblatt fur Veterinarmedizin. Reihe A 45, 577-579 (1998). <https://doi.org/10.1111/j.1439-0442.1998.tb00861.x>

Discussion

- Genomic PCR on the placenta is not sufficient to assert trophoblast contribution because the signal could be derived from mesoderm derivatives. The authors should be cautious on the issue of expanded potency (which is not defined!)

Response: We agree with the reviewer's concern. As a result, we have removed this section from the discussion in the revised manuscript.

- The discussion would benefit from describing expression of ERAS, PRMT6 and KLF2 in the early rabbit embryo.

Response: PRMT6 and KLF2 are expressed in rabbit morulae but are down-regulated in ICM/epiblast. In contrast, we were unable to detect ERAS transcripts in rabbit embryos (see Discussion pages 16, lines 462-471).

- It is somewhat misleading to compare the observed chimerism with that for mouse ES cells. The frequency of fetuses and term animals with detectable contributions may be similar, but the extent of colonization is clearly less than can be routinely obtained in mouse or rat.

Response: We agree with the reviewer's point. The sentence has been revised to clarify this comparison: "...comparable to those observed in mice following ESC injection." (see page 18, lines 528-529).

- Gene invalidation is not correct terminology

Response: We have replaced the term "invalidation" with "inactivation" in the revised manuscript (page 18, line 537).

Reviewer #2 (Remarks to the Author):

In the article "Viable chimeric rabbits with a high contribution of induced pluripotent stem cells", the authors have pioneered a novel method for generating chimeric rabbits by refining the culture medium and overexpressing specific genes. The study's findings are substantial in their potential to advance rabbit use in laboratory research. Although germline chimeric rabbits were not generated, the investigation is compelling nonetheless. However, prior to considering publication, I would like to see the following revisions:

1. As described in the manuscript, rabbit embryonic stem cells (ESCs) cultured in KF medium exhibit a primed state of pluripotency. However, this finding is intriguingly at odds with the observation that NaiveRep_KF cells express the naive-specific reporter EOS-TagBFP. To reconcile these seemingly contradictory results, the authors should provide further explanation or discussion in the text (Fig. 1g).

Response: The EOS-TagBFP reporter is driven by the ubiquitously active ETn early transposon promoter, which is coupled with the naïve state-specific distal enhancer of OCT4/POU5F1 (Hotta et al., 2009, 6: 370-376). The basal expression of TagBFP in NaiveRep_KF cells likely results from the activity of the ETn promoter. We have included this explanation in the revised manuscript, specifically in the legend for Figure 1.

2. For the four rounds of screening, it is notable that Screen 1 differs from the other three screenings (Screens 2a, 2b, and 3). Specifically, the cDNA library used in Screen 1 contains only 25 genes, whereas Screens 2a, 2b, and 3 utilize a library with 36 genes. Additionally, the multiplicity of infection (MOI) for Screen 1 is 50, whereas it is 15 for the other screens. These discrepancies warrant further explanation or discussion in the text to clarify their impact on the results.

Response: The three selected genes, KLF2, ERAS, and PRMT6, were included in the 25-gene library of Screen 1. None of the 11 additional genes in Screens 2a, 2b, and 3 were enriched during those screens. The higher MOI in Screen 1 (50) resulted in cell clones with 2 to 24 provirus integrations (mean = 9.3), which was too high for stringent enrichment in naïve pluripotency-promoting genes. However, we did observe an enrichment in the KLF2/ERAS/PRMT6 combination. To address this, we reduced the MOI to 15 for Screens 2a, 2b, and 3, which resulted in clones with 1 to 15 provirus integrations (mean = 4.8). These changes had minimal impact on the overall outcome of the screens, which we discuss further in the revised manuscript.

3. The statements "This indicates that transgene silencing shifted KEPI_VAL cells away from naïve-like pluripotency" and "Nevertheless, the Shield1-deprived cells continued to self-renew without any apparent differentiation" appear to be contradictory. The first statement suggests that transgene silencing leads to a loss of naïve-like pluripotency, while the second statement implies that the cells remained in a state of self-renewal without differentiating. This possible inconsistency should be addressed or clarified in the text.

Response: We thank the reviewer for pointing out this inconsistency. To clarify, we have reformulated the sentence as follows: "This indicates that degradation of the transgene product destabilized KEPI_VAL cell self-renewal in the naïve state of pluripotency but did not lead to visible differentiation for the duration of the experiment." This revision, which can be found on page 11 (lines 323-325) of the revised manuscript, hopefully resolves the potential contradiction.

4. The manuscript reports on the presence of CD75^{high} cells in KEPI#13_VAL+S and KEPI#28_VAL+S. Specifically, what is the percentage of CD75^{high} cells in these populations? Furthermore, are the percentages of CD75^{high} cells different between Condition I (24-hour culture) and Condition II (5-day culture)?

Response: The CD75^{high} cells do not form a distinct population (Fig. 6b). Instead, the expression of CD75 is detected as a continuum, ranging from undetectable to high levels. For cell injection, we selected the top 2.1% (KEPI#28) and 3.5% (KEPI#13) of cells with the highest CD75 expression (see

Figure 6b, Figure S8 and page 12 [lines 349-353] of the revised manuscript). This threshold was chosen primarily to ensure adequate cell numbers for culture and injection on the following day. Due to the scarcity of CD75^{high} cells after sorting, we were unable to analyze CD75 expression after a 24-hour culture period. However, after 5 days in culture, the percentage of CD75^{high} cells decreased to 41%, indicating the instability of CD75 expression. This reduction in the proportion of CD75^{high} cells correlates with the slight decline in the rate of chimera formation observed 5 days post-FACS compared to 24 hours post-FACS.

Studying the stability of CD75 expression would require a time-course experiment to reach a firm conclusion. However, as this question has no direct impact on the outcome of the chimera experiments (i.e. CD75^{high} cells are preferentially injected after 24 h post-FACS), we believe it is beyond the scope of the present study.

5. In Figure 7a, the electrophoretic image appears to show two distinct lines corresponding to muscles. Are these two lines representatives of different kind of muscles?

Response: We collected muscle samples from two different parts of the body: the anterior and posterior regions. In the revised manuscript, we have updated Figure 7a (Fig. S10 in the revised manuscript) to clearly label these as muscle #1 and #2 for clarity (see Newborn rabbit #1 in Fig. S9).

6. Can the GFP gene be detected in germline-related cells such as Sertoli cells, sperm, oocytes, or cumulus cells?

Response: We examined the gonads of our three adult female chimeras and observed strong colonization within the ovarian follicles. Additionally, we analyzed the progeny of these three female founders and confirmed germline transmission of the iPSC genome for two of them (the third gave no offspring). Notably, the rate of transmission was 100%, suggesting that the germline of these two chimeras is entirely derived from the iPSCs. These new findings are detailed on page 13 (lines 387-407) and illustrated in Figs. 7 and S12 of the revised manuscript.

7. For a stably established cell line, karyotype consistency is crucial. It is essential to demonstrate that the cells cultured in VAL with overexpressed KEP exhibit a stable chromosomal count and structure.

Response: We agree with the reviewer's concern. A karyotype analysis of KEPI#28_VAL+S cells is provided in Figure 5b of the revised manuscript (described on page 11, lines 307-308), demonstrating stable chromosomal count and structure.

8. It would be highly valuable to derive naive rabbit ESCs without genetic modification. Is it feasible to achieve this goal? A more in-depth discussion on the possibility and potential approaches for accomplishing this feat would be beneficial.

Response: We briefly addressed this possibility in the Discussion section of the revised manuscript (page 19, lines 538-541).

Reviewer #3 (Remarks to the Author):

These authors successfully generated viable chimeric rabbits using pluripotent stem cells. It is a great work and experiments were carefully performed. Importantly, they identified three genes—KLF2, PRMT6, and ERAS—that enable rabbit iPSCs to acquire molecular and functional features of the naïve state including the capacity to efficiently colonize preimplantation embryos. This work should be congratulated but the paper can be improved by providing the discussion on the possible germline transmission and potential application of this rabbit model for the research in biomedical field in the discussion section.

Response: We would like to thank the reviewer for his very positive assessment of our work. In the revised manuscript, we have addressed his/her request providing a substantial amount of new data. Specifically, we sacrificed the chimeric founders to examine chimerism rates across various organs using both quantitative genomic PCR and tissue section analysis. Additionally, we analyzed the progeny of two of the three female founders, observing germline transmission of the iPSC genome. These new findings are detailed on page 13-14 (lines 387-407) and illustrated in Figs. 7 and S12 of the revised manuscript. They are discussed in the Discussion section (pages 18, lines 522-541).

Reviewer #1 (Remarks to the Author):

Figure 7E shows the results of genomic PCR assay for GFP in foetuses from litters of the two female chimaeras. Every foetus (27/27) shows a strong signal for the GFP transgene. This is a quite extraordinary result that requires further explanation in two respects. Firstly, this result would mean that the iPSCs have completely displaced host germ cells. What proportion of oogonial cells in the ovaries of these two animals were GFP+ve (the text says numerous, not all; images are provided but no quantitation)?

Response: We examined ovary sections and observed that 8 oogonial cells out of 79 examined were GFP-negative (female A#2: 4/36; female A#6: 4/43). Thus, the iPSC-derived oogonial cells have not completely displaced host germ cells. These findings are described in the Results section (page 13).

Can the authors explain how could germline chimaerism reach 100% in animals that show very low contributions in most tissues (Rabbit 6 is <0.5% in all tissues except lung).

Response: This is indeed a surprising observation. We hypothesize that the KEP_VALGÖX reprogramming protocol primes CD75high cells toward the female germline, although they do not express markers of primordial germ cells. We are currently pursuing further studies to validate this hypothesis. However, we believe addressing this important question in full detail is beyond the scope of the present manuscript.

Secondly, even if all the oocytes are iPSC derived, they would not all be expected to transmit the GFP transgene due to segregation during meiosis. We are not told how many copies of the transgene are in the B19-GFP cells, but there would have to be several independent integrations to explain this result.

Response: Thank you for pointing this out. Indeed, the B19-GFP cells were engineered using a lentiviral vector with a multiplicity of infection (MOI) of 20. As demonstrated in Supplementary Figures 2c and 2d, this MOI resulted in a minimum of four independent proviral integrations in approximately 50% of the cells. Consistent results were obtained in the 3rd screening experiment, where a MOI of 15 produced an average of 4.8 +/- 2.0 proviral integrations per clone. These multiple proviral integrations were intentionally achieved to overcome silencing and ensure GFP expression across all cell lineages. This clarification has been added to the Discussion in the revised manuscript (page 17).

Furthermore, in such a case the copy numbers would vary between the offspring, which is not evident from the PCR gel. Please clarify and provide evidence to justify.

Response: We repeated the quantitative genomic PCR experiment using a reduced amount of template DNA (100 ng instead of 500 ng) and observed clear differences across the samples. The new PCR gels images are now included in Figure 7e and described on page 13.

Moreover, epifluorescence images are not definitive without showing non-transgenic foetuses in the same image and/or corroboration by GFP antibody staining as elsewhere in the paper. In addition, information should be provided whether all foetuses have normal anatomy and developmental staging.

Response: In the revised manuscript, we have included a non-transgenic foetus in the same image for comparison (see Figure 7f). We observed no appreciable differences in external anatomy and developmental stage between GFP fetuses and controls. This clarification has been added to the revised manuscript (page 13).

The transgenic foetuses were sacrificed for PCR analysis, precluding histological analysis with GFP antibody. Moreover, the chimeric founders had to be sacrificed to conduct histological analysis and to comply with the terms of our animal experimentation agreement. Consequently, a comprehensive assessment of F1 foetus development could not be performed and is beyond the scope of this manuscript.

Finally, there is a discrepancy between the number of germline foetuses declared in the text, 25, and the 27 samples shown in Fig 7E.

Response: Thank you for pointing this out. Our study reports 27 germline fetuses (see page 13)

The revised Introduction is much improved. The statement on lines 69-71 that “In mice, this has been achieved by reprogramming EpiSCs through the overexpression of specific transcription factors, including Klf4, Nr5a1, and Nr5a2, in culture conditions with LIF, MEK inhibitor PD0325901, and GSK3 inhibitor CHIR99021, known as 2iLIF” is somewhat misleading, however. Germline transmission was first shown for mouse ES cells by Bradley et al in 1984 and 2iLIF culture was developed for naive ES cells without use of transgenes. I suggest those points should be made before EpiSC reprogramming.

Response: We agree that the original statement was misleading, indeed. We have rephrased it to improve clarity and accuracy, as follows (page 3): “A critical requirement for generating germline chimeras is the ability of pluripotent cells to robustly self-renew in the naive state. In mice, this is typically achieved using culture conditions containing LIF, MEK inhibitor PD0325901, and GSK3 inhibitor CHIR99021, known as 2iLIF^{3,14,15}. Germline chimerism has also been achieved by reprogramming EpiSCs through the overexpression of specific transcription factors, including Klf4, Nr5a1, and Nr5a2, in 2iLIF culture conditions^{16,17}».

The Results text should state when first describing VALGoX medium that this is in MEF-conditioned medium (which they later show is required).

Response: We have clarified in the revised manuscript that VALGoX medium is used in MEF-conditioned medium. This is now explicitly stated on page 4 (top).

In Table 2 it is reported that chimaera A2 has 100% iPSC contribution by quantitative gPCR analysis, but the image of skin from that animal in Fig 7C shows a large number of GFP-ve cells. This difference should be discussed.

Response: The observed difference arises from the variations in the degree of chimerism across different parts of the body. Such variations were noted in several organs. For instance, as shown in Figure 6a, the rate of chimerism differs significantly between the right and left sides of the neural tube. We have addressed this discrepancy between Table 2 and Fig. 7c in the revised manuscript (see page 13).

Line 524: “These CD75^{high} cells have reactivated the second X chromosome” is over-interpreting the histone modification staining as raised in the previous review.

Response: Thank you for pointing this out. We have clarified in the revised manuscript on page 17: “These CD75^{high} cells lack the typical histone modification of inactive 2nd X chromosome”.

In Table 1 there is some mislabelling in the Sex and Name column

Response: We have corrected it in the revised ms.

Reviewer #1:

Transgene transmission: They provide a new gPCR analysis using reduced template DNA that shows different intensities of GFP band. However, there is no loading control provided for this analysis, nor any copy number titration. More fundamentally, the number of integrations in the B19 line is assumed to be multiple, but this is not evidenced. Integration site sequence analysis should be performed to quantify the number of integrations and to demonstrate inheritance of different subsets. I recognise that I did not specify this assay in my review, but it is a reasonable expectation. There should not be a technical problem in performing integration analyses on the DNA samples the authors have in hand. The observation of 100% transgene transmission is so unusual and the proof of germline competence for a non-rodent embryonic stem cell so significant for the wider field, that there should be no room left for doubt.

To address the reviewer's concerns, we conducted an in-depth analysis of GFP transgene transmission. Using a ligation-mediated PCR assay on the KEPi#28 iPSCs that produced the germline chimeras, we identified five independent integration sites distributed across five different chromosomes. We then applied the same approach to 11 of the 27 F1 fetuses (selected based on DNA availability) and demonstrated that each integration site was transmitted to one or more fetuses. Overall, 8 of these 11 F1 fetuses (73%) were found to be transgenic. We believe that the three fetuses testing negative likely harbor integration sites that were not detected with SacI- and KpnI-digested DNA post-amplification. Unfortunately, due to limited DNA availability, we were unable to perform additional analyses using alternative restriction enzymes.

These new data are described on pages 13 and 14 (Results), pages 25-27 (Material and Methods), page 38 (Table 3), and page 42 (legend Figure 7).